# A Unifying Framework for Gradient Aggregation in Multi-Objective Optimization

## Abstract

Many machine learning problems involve multiple inherent trade-offs that are best addressed by gradient-based multi-objective optimization (MOO) algorithms. Existing methods are often proposed with various motivations, analyzed case by case, and differ algorithmically in how the component gradients are aggregated at each step. In this work, we develop a unifying framework for gradient aggregation in MOO, establishing (optimal) rates of convergence to Pareto stationarity—the standard measure of performance in MOO. Central to our analysis is a sufficient alignment condition, from which we derive a theorem showing that non-conflicting directions, when chosen within the convex hull of gradients, form a fundamental sufficient condition for convergence. We further show that feasibility can be ensured through projection onto the dual cone, broadening the scope of methods that admit convergence guarantees. In parallel, we present a primal optimization perspective of gradient aggregation that encompasses established algorithms, clarifies their theoretical relationships, and enables the design of new variants. As an illustration, we introduce capped MGDA, derived from a CVaR-based formulation, and demonstrate its robustness in adversarial federated learning. Finally, we validate our theory through experiments on synthetic problems and practical fairness benchmarks.

## 1 Introduction

Many contemporary problems in machine learning are inherently multi-objective, requiring a balance between multiple, often competing, performance criteria. This tension is evident across diverse applications, from ensuring fairness alongside accuracy in classification systems, to balancing heterogeneous clients' performances in federated learning (FL), and to jointly mastering different tasks with a shared model in multi-task learning (MTL). To address such challenges at the scale of modern deep learning, gradient-based MOO methods have become indispensable, offering scalability to high-dimensional models and seamless integration with existing training pipelines. In these methods, the key algorithmic challenge is to determine, at each iteration, an effective update direction $\mathbf{d}$ synthesized from the component gradients, that can guide learning across competing objectives.

Recent work on gradient aggregation in MOO spans a range of algorithms—e.g., MGDA (Désidéri, 2012), Nash-MTL (Navon et al., 2022), FairGrad (Ban & Ji, 2024a), and UPGrad (Quinton & Rey, 2024), among others—each proposing a particular rule for constructing $\mathbf{d}$ from component gradients. These methods were developed under disparate motivations and, when available, their convergence analyses are established case by case, tied to method-specific assumptions and proofs. As a result, while these methods have provided valuable insights, there is still no general framework to explain what properties of an update direction ensure convergence to Pareto stationarity or how these different methods are connected. This gap highlights the need for a unifying theory that clarifies the conditions for convergence and offers a principled basis for designing new aggregation schemes.

In this work, we develop a general theoretical framework for gradient-based MOO. Our first main result (Theorem 1 and Corollary 1) establishes a broad *alignment condition* (A) on $\mathbf{d}_t$ that guarantees convergence to Pareto stationarity. This versatile result makes Corollary 1 pivotal and serves as the cornerstone of our analysis. Building on it, we derive Theorem 2, which specializes condition (A) to the convex hull and non-conflicting requirements, thereby explaining the success of prominent non-conflicting aggregation rules. We further show that feasibility can be restored by projection

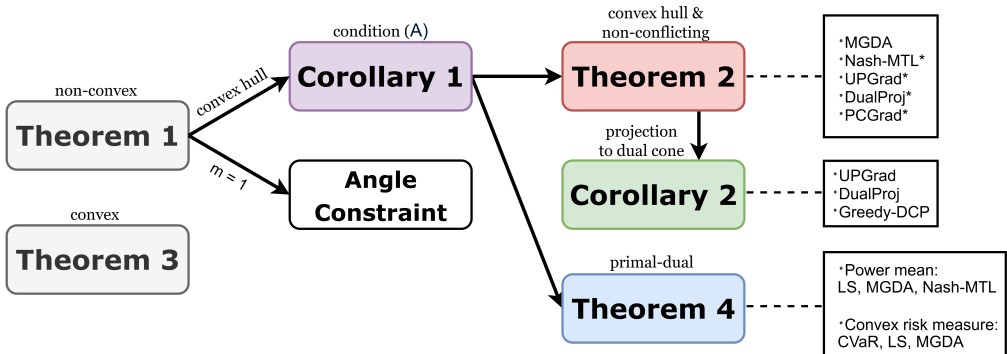

Figure 1: An overview of the relationships between our key theoretical results. Our pivotal result, Theorem 1 and Corollary 1, establishes a broadly applicable convergence guarantee that requires condition (A). From this result, we then derive Theorem 2, Corollary 2, and Theorem 4, which provide more insightful and readily verifiable criteria for a broad class of MOO aggregation methods.

onto the dual cone, leading to Corollary 2. In parallel, we introduce the (primal) optimization subproblem perspective of gradient aggregation (12), and establish sufficient conditions under which the resulting aggregation is a conic combination of gradients, ensuring convergence (Theorem 4). This perspective subsumes existing formulations and provides a principled recipe for designing new ones. Together, these results form a coherent unifying framework that simplifies theoretical analysis, clarifies prior work, and opens new design possibilities. We summarize our contributions below:

- We establish a general alignment criterion for gradient-based MOO, yielding a broadly applicable template for analyzing convergence.

- We uncover the theoretical importance of non-conflicting directions in MOO as a fundamental condition leading to convergence.

- We study the primal optimization subproblem formulation of gradient aggregation, providing sufficient conditions for the resulting aggregation to be in the conic hull and converge, subsuming several existing methods (e.g., LS, MGDA, Nash-MTL) and clarifying their relationships.

- We design and analyze a novel method, capped MGDA, derived from a CVaR-based primal formulation, illustrating our framework's ability to generate new aggregations.

- We validate our theoretical results through experiments on synthetic and fairness benchmarks, and on adversarial federated learning, demonstrating the robustness of capped MGDA.

## 2 RELATED WORKS

Multi-objective optimization (MOO) and Pareto solutions have been extensively studied, with classical approaches such as evolutionary algorithms (Deb et al., 2002). However, modern ML problems are large-scale and differentiable, making gradient-based methods more appropriate. Therefore, in this work we focus on gradient-based MOO, in particular, multi-objective gradient aggregation.

**Gradient-based MOO.** Gradient-based MOO optimizes multiple objectives using gradient information. A foundational algorithm in this regard is MGDA (Mukai, 1980; Fliege & Svaiter, 2000; Désidéri, 2012), which computes a *non-conflicting* direction by solving for the minimum-norm element in the convex hull of gradients. Fliege et al. (2019) provided a detailed convergence analysis of MGDA. Subsequent works (e.g., Fliege et al. 2009; O. Montonen & Mäkelä 2018; Tanabe et al. 2019; Assunção et al. 2021; Tanabe et al. 2023) have also extended classical single-objective methods to the multi-objective setting. Another important line of research studies stochastic variants of MGDA (Mercier et al., 2018; Liu & Vicente, 2021; Zhou et al., 2022; Fernando et al., 2023; Chen et al., 2023; Xiao et al., 2023), motivated by their practical relevance in machine learning, particularly for mini-batch training of deep neural networks.

**Multi-task Learning (MTL) and Multi-Objective Gradient Aggregation (MOGA).** MTL aims to train a single model that performs well across multiple tasks. Sener & Koltun (2018) first cast MTL as a multi-objective optimization problem and applied MGDA to address it. Since then, a rich

line of research in MTL has proposed general-purpose multi-objective gradient aggregation methods, focusing on novel gradient aggregation schemes to mitigate task conflicts. Examples include PCGrad (Yu et al., 2020), which projects each gradient onto the normal plane of others; CAGrad (Liu et al., 2021a), which balances average and worst-case objectives by constraining the search region; and Nash-MTL (Navon et al., 2022), which frames MTL as a bargaining game. Other approaches include IMTL-G (Liu et al., 2021b) and FairGrad (Ban & Ji, 2024b), among others.

**Non-conflicting direction and the Dual cone.** The notion of a non-conflicting update direction has appeared in various MOO-related works (Désidéri, 2012; Yu et al., 2020; Liu et al., 2021a), though often without sufficient formalization or emphasis. Recent studies clarified that this criterion corresponds to a dual cone constraint over the gradients $\mathbf{g}_k$, which can be explicitly enforced to guarantee conflict-free updates (Hwang & Lim, 2024; Quinton & Rey, 2024). While Quinton & Rey (2024) acknowledge the relevance of non-conflicting directions and propose projection onto the dual cone to ensure them, they do not investigate its theoretical significance. In contrast, our work rigorously establishes non-conflicting as a unifying sufficient condition for convergence to Pareto stationarity (see Theorem 2). We show that non-conflicting is not merely a preference, but a fundamental condition for convergence guarantees—something not recognized in prior work.

## 3 PRELIMINARIES

This section reviews the concepts of Pareto optimality, Pareto stationarity, a measure for quantifying the latter, and two key cones associated with the Jacobian matrix in multi-objective optimization.

### 3.1 MULTI-OBJECTIVE OPTIMIZATION (MOO)

In mathematical terms, a Multi-Objective Optimization (MOO) problem can be written as:

$$\min_{\mathbf{w} \in \mathbb{R}^d} \mathbf{f}(\mathbf{w}), \quad \text{where} \quad \mathbf{f}(\mathbf{w}) := (f_1(\mathbf{w}), f_2(\mathbf{w}), \ldots, f_m(\mathbf{w})) \tag{1}$$

and the minimum is defined w.r.t. the *partial* ordering:

$$\mathbf{f}(\mathbf{w}) \leq \mathbf{f}(\mathbf{z}) \iff \forall i = 1, \ldots, m, \ f_i(\mathbf{w}) \leq f_i(\mathbf{z}). \tag{2}$$

Unlike single-objective optimization, with multiple objectives it is possible that

$$\mathbf{f}(\mathbf{w}) \not\leq \mathbf{f}(\mathbf{z}) \text{ and } \mathbf{f}(\mathbf{z}) \not\leq \mathbf{f}(\mathbf{w}), \tag{3}$$

in which case we say $\mathbf{w}$ and $\mathbf{z}$ are not comparable. As a result, a MOO problem typically admits a set of optimal solutions (a.k.a. *Pareto Optimal*), whose objective values form the *Pareto front*.

### 3.2 PARETO OPTIMALITY AND PARETO STATIONARITY

**Definition 1** (Pareto Optimality)**.** *We call* $\mathbf{w}^*$ *a* Pareto optimal *solution of* (1) *if its objective value* $\mathbf{f}(\mathbf{w}^*)$ *is a minimum element w.r.t. the partial ordering in* (2)*; equivalently,*

$$\forall \mathbf{w}, \ \mathbf{f}(\mathbf{w}) \leq \mathbf{f}(\mathbf{w}^*) \implies \mathbf{f}(\mathbf{w}) = \mathbf{f}(\mathbf{w}^*). \tag{4}$$

In other words, it is not possible to improve *any* component objective in $\mathbf{f}(\mathbf{w}^*)$ without compromising *some* other objective. Similarly, we call $\mathbf{w}^*$ *weakly* Pareto optimal if it is not possible to improve *all* objectives in $\mathbf{f}(\mathbf{w}^*)$, i.e., there does not exist $\mathbf{w}$ such that $\mathbf{f}(\mathbf{w}) < \mathbf{f}(\mathbf{w}^*)$.

Next, we recall the concept of *Pareto Stationarity* (also referred to as *Pareto Criticality*), which is the first-order necessary condition for Pareto optimality.

**Definition 2** (Pareto Stationarity)**.** *We call* $\mathbf{w}^*$ *Pareto stationary (PS) iff*

$$\mathbf{0} \in \text{conv}\{\nabla f_1(\mathbf{w}^*), \cdots, \nabla f_m(\mathbf{w}^*)\}, \tag{5}$$

*i.e., there exists some* $\boldsymbol{\lambda} \in \Delta$ *(the probability simplex) such that* $\sum_{i=1}^m \lambda_i \nabla f_i(\mathbf{w}^*) = \mathbf{0}$.

The relevance of Pareto stationarity is captured in the following lemma:

**Lemma 1** (e.g., Mukai 1980, Theorem 1)**.** *Any Pareto optimal solution is Pareto stationary. Conversely, if all functions are convex (resp., strictly convex), then any Pareto stationary solution is weakly Pareto optimal (resp., Pareto optimal).*

**Measure of Pareto Stationarity.** To quantify the degree of Pareto stationarity, we recall the following metric (e.g., Mukai 1980; Chen et al. 2023; Zhang et al. 2025):

$$\gamma(\mathbf{w}) = \gamma_\mathbf{f}(\mathbf{w}) := \min_{\boldsymbol{\lambda} \in \Delta} \|J_\mathbf{f}(\mathbf{w})\boldsymbol{\lambda}\|, \quad \text{where} \quad J_\mathbf{f}(\mathbf{w}) := [\nabla f_1(\mathbf{w}), \dots, \nabla f_m(\mathbf{w})]. \tag{6}$$

Clearly, $\gamma(\mathbf{w}) = 0$ iff $\mathbf{w}$ is Pareto stationary. When $m = 1$ (single-objective), $\gamma(\mathbf{w}) = \|\nabla f(\mathbf{w})\|$ is the standard gradient norm widely used in analyzing gradient descent for nonconvex functions. This measure $\gamma(\mathbf{w})$ is continuous (assuming $\mathbf{f}$ is continuously differentiable). Therefore, when $\mathbf{w}_t \to \mathbf{w}_*$ and $\gamma(\mathbf{w}_t) \to 0$, we immediately know that the limit $\mathbf{w}_*$ must be Pareto stationary since $\gamma(\mathbf{w}_*) = 0$.

We introduce two cones in $\mathbb{R}^d$ that are related to a matrix $J \in \mathbb{R}^{d \times m}$:

$$\text{cone}\, J := \{\mathbf{d} : \mathbf{d} = J\boldsymbol{\mu}, \boldsymbol{\mu} \geq \mathbf{0}\}, \qquad \text{cone}^* J := \{\mathbf{d} : J^\top \mathbf{d} \geq \mathbf{0}\}. \tag{7}$$

Setting $J$ to be the (transposed) Jacobian $J_\mathbf{f}(\mathbf{w})$ at each iteration, the two cones represent two natural conditions on the update direction $\mathbf{d}$:

- $\text{cone}\, J_\mathbf{f}(\mathbf{w})$ consists of directions that are conic combinations of the component gradients;
- $\text{cone}^* J_\mathbf{f}(\mathbf{w})$ consists of directions that are *non-conflicting* with each component gradient.

We note that a direction $\mathbf{d} \in \text{cone}\, J$ can be normalized to lie in $\text{conv}\, J$, and the normalization constant can be absorbed into the step size.

# 4 A Unified Proof and Algorithmic Framework for MOO

In this section we consider the following general update for MOO:

$$\mathbf{w}_{t+1} = \mathbf{w}_t - \eta_t \mathbf{d}_t, \tag{8}$$

where $\eta_t > 0$ is the step size and $\mathbf{d}_t$ is the update direction. We will first present a general theorem for analyzing the progress of the above update. Then, we derive immediate consequences of our framework, illustrate the construction of the update direction, and detail some examples.

## 4.1 What directions lead to convergence

We recall that a function $F$ is $L$-smooth if its gradient $\nabla F$ is $L$-Lipschitz continuous, which is a widely adopted assumption in the analysis of gradient-based methods (Nesterov, 2018).

Our first result is a slight generalization of the well-known result on feasible directions (e.g., Zoutendijk, 1976, §14.2).

**Theorem 1** (Sufficient Alignment Condition)**.** *Suppose there exists an $L$-smooth function $F : \mathbb{R}^d \to \mathbb{R}_+$ such that the directions $\mathbf{d}_t$ satisfy:*

$$\langle \mathbf{d}_t, \nabla F(\mathbf{w}_t) \rangle \geq c_t \Gamma_t \|\mathbf{d}_t\|, \qquad \text{with } c_t \geq 0. \tag{9}$$

*With suitably chosen step size $\eta_t$ (so that (59) in Appendix B holds; for instance, when $\eta_t = \frac{c_t \Gamma_t}{L \|\mathbf{d}_t\|}$), if $c_t \geq c > 0$, then $\sum_t \Gamma_t^2 \leq \frac{2LF(\mathbf{w}_0)}{c^2}$. In particular, $\min_{t \leq T} \Gamma_t \leq \sqrt{\frac{2LF(\mathbf{w}_0)}{c^2 T}}$ and $\lim_{t \to \infty} \Gamma_t = 0$.*

Despite the simplicity of its proof, Theorem 1 is surprisingly general: There is little restriction on how the direction $\mathbf{d}_t$ or the quantity of interest $\Gamma_t$ is chosen. In the single-objective setting, letting $\Gamma_t = \|\nabla F(\mathbf{w}_t)\|$ we reduce to the well-known angle constraint in the method of feasible directions (e.g., Zoutendijk, 1976, §14.2):

$$\frac{\langle \mathbf{d}_t, \nabla F(\mathbf{w}_t) \rangle}{\|\mathbf{d}_t\| \cdot \|\nabla F(\mathbf{w}_t)\|} \geq c_t > 0. \tag{10}$$

In our multi-objective setting, the function $F$ serves as a (proof) surrogate: we use it to prove the convergence of $\Gamma_t := \gamma(\mathbf{w}_t)$, the measure of Pareto stationarity. Often we can simply choose $F$ to be linear combinations (or even just the sum) of the component functions $f_k$ in MOO.

Interestingly and most importantly, upon setting[1] $\Gamma_t = \|\mathbf{d}_t\|$, condition (9) simplifies to

$$\langle \mathbf{d}_t, \nabla F(\mathbf{w}_t) \rangle \geq c_t \|\mathbf{d}_t\|^2 \geq 0. \tag{A}$$

Theorem 1 then gives conditions on when $\mathbf{d}_t \to 0$. To relate this back to the measure of Pareto stationarity (i.e., $\gamma(\mathbf{w}_t)$ in (6)), we need only restrict $\mathbf{d}_t$ to the convex hull of the component gradients, so that $\|\mathbf{d}_t\| \geq \gamma(\mathbf{w}_t)$ holds trivially. We summarize this observation in a corollary since it highlights the convenience of searching the direction $\mathbf{d}_t$ in the convex hull of component gradients:

**Corollary 1.** *If $\mathbf{d}_t \in \mathrm{conv}(J_{\mathbf{f}}(\mathbf{w}_t))$ and condition (A) holds with $c_t \geq c > 0$. Then, with (constant) step size $\eta_t = \frac{c}{L}$, we have $\min_{t \leq T} \gamma(\mathbf{w}_t) \leq \sqrt{\frac{2LF(\mathbf{w}_0)}{c^2 T}}$.*

In other words, we approach Pareto stationarity at the rate of $O(1/\sqrt{t})$, which in general is optimal (even for the single-objective case). We note that a larger choice of $F$ makes it easier to satisfy the condition (A), but this also increases the constants $L$ and $F(\mathbf{w}_0)$ in the rate of convergence.

Based on Corollary 1 we now present our second result, a surprisingly simple and yet effective sufficient condition for convergence to Pareto stationarity.

**Theorem 2** (Convergence of Non-Conflicting Directions). *If the direction $\mathbf{d}_t \in \mathrm{conv}(J_{\mathbf{f}}(\mathbf{w}_t))$ and $\mathbf{d}_t \in \mathrm{cone}^*(J_{\mathbf{f}}(\mathbf{w}_t))$ (i.e., non-conflicting), then condition (A) and hence Corollary 1 holds with $c_t \equiv 1$ and $F = \sum_k f_k$.*

Quite remarkably, the convex hull and the non-conflicting conditions are easy to check (and construct, as we shall see), and together they already imply the (optimal) $O(1/\sqrt{t})$ rate of convergence to Pareto stationarity. Interestingly, we can obtain a non-conflicting direction through projection:

**Proposition 1.** *Let $\mathbf{q} \in \mathrm{cone}(J)$, i.e., $\mathbf{q} = J\boldsymbol{\mu}$ for some $\boldsymbol{\mu} \geq 0$. Then, $\mathbf{d} := \mathrm{P}_{\mathrm{cone}^*(J)}(\mathbf{q}) \in \mathrm{cone}^*(J) \cap \mathrm{cone}(J)$. In particular, $\mathbf{d} = J\boldsymbol{\nu}$ for some $\boldsymbol{\nu} \geq 0$ such that $\|\boldsymbol{\nu}\|_1 \geq \|\boldsymbol{\mu}\|_1$.*

Thus, surprisingly but conveniently, from an algorithmic point of view, it suffices to choose a (pre)direction $\mathbf{q}_t$ from the convex hull of the component gradients:

**Corollary 2.** *Let $\mathbf{d}_t = \mathrm{P}_{\mathrm{cone}^*(J_t)}(\mathbf{q}_t)$ where $\mathbf{q}_t \in \mathrm{conv}(J_t)$ and $J_t := J_{\mathbf{f}}(\mathbf{w}_t)$, then (A) and hence Corollary 1 holds with $c_t \geq 1$.*

Furthermore, we observe that condition (A) is stable under convex combinations, namely that if each direction $\mathbf{d}_j$ satisfies (A), then so does any of their convex combinations. UPGrad (Quinton & Rey, 2024) is an example method that takes the average of these projected directions. See also Appendix A.2.2 for another novel variant whose convergence follows directly from Corollary 2.

**Convex case.** When the component functions $f_k$ are convex, by slightly strengthening the non-conflicting property of the direction $\mathbf{d}_t$, we can establish an $O(\frac{1}{t})$ convergence rate in terms of the (aggregated) function value. See Appendix B for detailed proof and discussion.

**Theorem 3** (Convergence under Monotone Descent). *Suppose each objective $f_k$ is L-smooth, convex and bounded from below. Choose $\mathbf{d}_t \in \mathrm{conv}(J_{\mathbf{f}}(\mathbf{w}_t))$ (and step size $\eta_t \equiv \eta \leq \frac{1}{L}$) such that the function values $\{\mathbf{f}(\mathbf{w}_t)\}$ monotonically decrease. Then, there exists $\boldsymbol{\lambda} \in \Delta$ such that the iterates $\mathbf{w}_t$ defined in (8) satisfy: for any $\mathbf{w}$,*

$$\langle \boldsymbol{\lambda}, \mathbf{f}(\mathbf{w}_t) \rangle - \langle \boldsymbol{\lambda}, \mathbf{f}(\mathbf{w}) \rangle \leq \frac{1}{2\eta t} \|\mathbf{w}_0 - \mathbf{w}\|^2. \tag{11}$$

In particular, choosing $\mathbf{w}_* = \mathrm{argmin}_{\mathbf{w}} \langle \boldsymbol{\lambda}, \mathbf{f}(\mathbf{w}) \rangle$, which is weakly Pareto optimal under convexity and Pareto optimal under strict convexity, we conclude that the iterates $\mathbf{w}_t$ converge at rate of $O(1/t)$ (in terms of function value). This result generalizes that of Fliege et al. (2019), from the particular method MGDA to any direction $\mathbf{d}_t \in \mathrm{conv}(J_{\mathbf{f}}(\mathbf{w}_t))$ that lies in the interior of $\mathrm{cone}^*(J_{\mathbf{f}}(\mathbf{w}_t))$ (which guarantees descending). Needless to say, when $m = 1$ (single-objective), Theorem 3 reduces to the well-known result of gradient descent.

In the next subsection, we present another way to construct the direction $\mathbf{d}_t$, followed by some examples that recover existing algorithms and uncover new variants.

---

[1] For the purpose of understanding the main results in this paper, one may simply take $\Gamma_t = \|\mathbf{d}_t\|$ in Theorem 1. This choice upper-bounds the measure of Pareto stationarity $\gamma(\mathbf{w}_t)$ and leads to condition (A), which is easier to verify in practice. Other choices (e.g., fractional powers of $\|\mathbf{d}_t\|$) are also possible, but setting $\Gamma_t = \|\mathbf{d}_t\|$ is sufficient for establishing all novel theoretical results presented in this work.

Table 1: Summary of gradient aggregation methods (non-exhaustive). $s(\mathbf{x})$, $r$, and $\mathbf{q}_t$ follow (12). Row colors match Figure 1, indicating the most specific theorem or corollary applicable to ensure convergence for each method, albeit more general ones may also apply. Details are in Appendix A.

| | $s(\mathbf{x})$ | $r$ | $\mathbf{q}_t$ |
|---|---|---|---|
| LS (Uniform) | $-\frac{\mathbf{1}^\top \mathbf{x}}{m}$ | $\frac{1}{2}\|\cdot\|^2$ | $J(\frac{\mathbf{1}}{m})$ |
| MGDA (Désidéri, 2012) | $\max_k(-x_k)$ | $\frac{1}{2}\|\cdot\|^2$ | $J\boldsymbol{\lambda}_{\mathrm{dual}_1}$ |
| Nash-MTL (Navon et al., 2022) | $-(\prod_k x_k)^{1/m}$ | $\frac{1}{2}\|\cdot\|^2$ | $J\boldsymbol{\lambda}_{\mathrm{eq}}$ |
| DualProj (Lopez-Paz & Ranzato, 2017) | $-\boldsymbol{\alpha}^\top \mathbf{x}$ | $\frac{1}{2}\|\cdot\|^2$ | $J\boldsymbol{\alpha}$ |
| UPGrad (Quinton & Rey, 2024) | $-(\frac{1}{m}\sum_k \boldsymbol{\alpha}_k)^\top \mathbf{x}$ | $\frac{1}{2}\|\cdot\|^2$ | $J(\frac{1}{m}\sum_k \boldsymbol{\alpha}_k)$ |
| CAGrad (Liu et al., 2021a) | $\max_k(-x_k)$ | $\iota_{\mathcal{B}_\epsilon(\mathbf{g}_0)}(\cdot)$ | $\mathbf{g}_0 + \epsilon J\boldsymbol{\lambda}_{\mathrm{dual}_2}$ |
| IMTL-G (Liu et al., 2021b) | $\|\mathbf{x} - \mathbf{n}\|^2$ | $\mu\|\cdot\|^2, \mu \downarrow 0$ | $(J^\dagger)^\top \mathbf{n}$ |
| Capped MGDA (new) | $\mathrm{CVaR}_\epsilon(\mathbf{x})$ | $\frac{1}{2}\|\cdot\|^2$ | $J\boldsymbol{\lambda}_{\mathrm{dual}_3}$ |
| Greedy-DCP (new) | $\min_k -\boldsymbol{\alpha}_k^\top \mathbf{x}$ | $\frac{1}{2}\|\cdot\|^2$ | $J\boldsymbol{\alpha}_K$ |

## 4.2 Constructing the update direction in the conic hull

We now show how to construct the update (pre-)direction in the conic hull of the component gradients so that (A) holds directly. Our construction is based on the optimization subproblem:

$$\mathbf{q}_t = \underset{\mathbf{q}}{\mathrm{argmin}} \; s(J_t^\top \mathbf{q}) + r(\|\mathbf{q}\|), \quad \text{where} \quad J_t := J_{\mathbf{f}}(\mathbf{w}_t) = [\nabla f_1(\mathbf{w}_t), \dots, \nabla f_m(\mathbf{w}_t)], \quad (12)$$

with $s : \mathbb{R}^m \to \mathbb{R}$ and $r : \mathbb{R}_+ \to \mathbb{R}$. W.l.o.g. we assume $s(\mathbf{0}) = r(0) = 0$. When $s$ and $r$ are convex, using the Fenchel conjugate $s^*(\boldsymbol{\lambda}) := \max_{\mathbf{w}} \langle \boldsymbol{\lambda}, \mathbf{w} \rangle - s(\mathbf{w})$, we derive the dual of (12):

$$\min_{\mathbf{q}} \; s(J_t^\top \mathbf{q}) + r(\|\mathbf{q}\|) = \min_{\mathbf{q}} \max_{\boldsymbol{\lambda}} \langle -\boldsymbol{\lambda}, J_t^\top \mathbf{q} \rangle - s^*(-\boldsymbol{\lambda}) + r(\|\mathbf{q}\|) \quad (13)$$

$$= \max_{\boldsymbol{\lambda}} \min_{\mathbf{q}} \langle J_t \boldsymbol{\lambda}, -\mathbf{q} \rangle - s^*(-\boldsymbol{\lambda}) + r(\|\mathbf{q}\|) \quad (14)$$

$$= -\min_{\boldsymbol{\lambda}} s^*(-\boldsymbol{\lambda}) + r^*(\|J_t \boldsymbol{\lambda}\|), \quad (15)$$

where $\mathbf{q} = \beta J_t \boldsymbol{\lambda}$ and $\beta = \frac{\nabla r^*(\|J_t \boldsymbol{\lambda}\|)}{\|J_t \boldsymbol{\lambda}\|} \geq 0$. When $s$ is also decreasing, we have $-\boldsymbol{\lambda} \leq \mathbf{0}$ and hence $\mathbf{q}_t \in \mathrm{cone}(J_t)$.

However, the direction $\mathbf{q}_t$ constructed above need not be non-conflicting (example will follow). Instead, we can directly establish the condition (A). For simplicity, let us assume $r(\|\mathbf{q}\|) \geq \frac{1}{2}\|\mathbf{q}\|^2$. From the optimality of $\mathbf{q}_t$ in (12):

$$0 = s(\mathbf{0}) + r(\|\mathbf{0}\|) \geq s(J^\top \mathbf{q}_t) + \frac{1}{2}\|\mathbf{q}_t\|^2, \quad i.e., \quad -s(J^\top \mathbf{q}_t) \geq \frac{1}{2}\|\mathbf{q}_t\|^2. \quad (16)$$

Then, to establish (A), we apply the convexity of $s$:

$$-s(J^\top \mathbf{q}_t) \leq -s(\mathbf{0}) + \langle -\nabla s(\mathbf{0}), J^\top \mathbf{q}_t \rangle = \langle -J_t \nabla s(\mathbf{0}), \mathbf{q}_t \rangle = \langle \nabla F(\mathbf{w}_t), \mathbf{q}_t \rangle, \quad (17)$$

upon choosing $F(\mathbf{w}) = \langle -\nabla s(\mathbf{0}), \mathbf{f}(\mathbf{w}) \rangle$. Thus, we have obtained (A) with $c_t = \frac{1}{2}$.

Let us summarize the above discussion in the following theorem.

**Theorem 4** (Subproblem-Based Construction of Convergent Directions). *Consider the optimization subproblem* (12). *Suppose $s$ is decreasing convex and $r(\rho) \geq \frac{1}{2}\rho^2$ is convex. Then, (I) the solution $\mathbf{q}_t$ to* (12) *lies in* $\mathrm{cone}(J_t)$ *with its normalized direction $\mathbf{d}_t$ lie in* $\mathrm{conv}(J_t)$, *and (II) $\mathbf{d}_t$ satisfies condition* (A) *with $c_t = \frac{1}{2}\beta_t \|\boldsymbol{\lambda}_t\|_1$ and $F = \langle -\nabla s(\mathbf{0}), \mathbf{f} \rangle$.*

Therefore, if we can show $c_t \geq c$ for some positive constant $c$, then Corollary 1 holds and the $O(1/\sqrt{t})$ rate of convergence to Pareto stationarity follows.

## 4.3 Examples: Old and New

In this section, we illustrate how the subproblem formulation (12) unifies a broad class of existing gradient aggregation methods, and even leads to the discovery of new ones (e.g., Capped MGDA) that are easy to implement and come with automatic convergence guarantees, thanks to Theorem 4.

### 4.3.1 POWER MEAN

We are now ready to present some examples. Let us first consider the *power mean*:

$$s(\mathbf{x}) = -\left(\tfrac{1}{m} \sum_k x_k^p\right)^{1/p}, \quad \text{where} \quad p \leq 1. \tag{18}$$

A similar formulation appeared in Ban & Ji (2024a) [see Appendix A.1.4 for details].

Restricted to $\mathbb{R}_+^m$, $s$ is decreasing and convex, with Fenchel conjugate

$$s^*(-\boldsymbol{\lambda}) = \begin{cases} 0, & \text{if } \left(\tfrac{1}{m} \sum_k \lambda_k^q\right)^{1/q} \geq 1, \boldsymbol{\lambda} \geq \mathbf{0} \\ \infty, & \text{otherwise} \end{cases}, \quad \text{where } 1/p + 1/q = 1. \tag{19}$$

According to the reverse Hölder's inequality, the optimal $\lambda_k \propto x_k^{p/q}$.

Setting $p$ differently allows us to recover some existing algorithms:

- $p = 1$: this amounts to linear scalarization with uniform weights, i.e., $s(\mathbf{x}) = -\tfrac{1}{m} \sum_k x_k$.
- $p \to 0$ (the limiting case): this corresponds to Nash-MTL (Navon et al., 2022), where $s(\mathbf{x}) = -(\prod_k x_k)^{1/m}$ is the geometric mean and (17) reduces simply to the arithmetic-geometric inequality (with $F = \tfrac{1}{m} \sum_k f_k$).
- $p \to -\infty$: this corresponds to MGDA (Désidéri, 2012; Mukai, 1980; Fliege & Svaiter, 2000), where $s(\mathbf{x}) = -\min_k x_k$.
- $p = -1$: this has been explored by[2] FairGrad (Ban & Ji, 2024a) and PIVRG (Qin et al., 2025).

**Convergence of power-mean-based directions.** With $r(\rho) = \tfrac{1}{2}\rho^2$, we have $\beta_t \equiv 1$ in Theorem 4. Since $q \leq 1$, we know $\|\boldsymbol{\lambda}\|_1 \geq \left(\tfrac{1}{m} \sum_k \lambda_k^q\right)^{1/q} \geq 1$ and hence $c_t \geq \tfrac{1}{2}$ in Theorem 4. Applying Corollary 1 we at once obtain the $O(1/\sqrt{t})$ rate of convergence to Pareto stationarity for all $p$.

### 4.3.2 CONVEX RISK MEASURES

Next, we choose $s$ from the family of *convex risk measures* (Föllmer & Schied, 2002), namely that $s$ is convex, decreasing and translation invariant:

$$\forall c \in \mathbb{R}, \ \ s(\mathbf{x} + c) = s(\mathbf{x}) - c. \tag{20}$$

The last two conditions ensure that the domain of the conjugate function

$$s^*(-\boldsymbol{\lambda}) = \max_{\mathbf{x},c} \langle -\boldsymbol{\lambda}, \mathbf{x} + c \rangle - s(\mathbf{x} + c) = \max_{\mathbf{x},c} \langle -\boldsymbol{\lambda}, \mathbf{x} \rangle - s(\mathbf{x}) + c(1 - \langle \boldsymbol{\lambda}, \mathbf{1} \rangle) \tag{21}$$

is restricted so that $\boldsymbol{\lambda} \in \Delta$ (the simplex). The power mean (18) with $p = 1$ and $p = -\infty$ are convex risk measures, while other values of $p$ are not.

**Capped MGDA via CVaR.** Another widely-used convex risk measure is the *Conditional Value-at-Risk (CVaR, Rockafellar & Uryasev 2000)*:

$$s(\mathbf{x}) = \mathrm{CVaR}_\epsilon(\mathbf{x}) = \min_\alpha \left\{ \alpha + \frac{1}{\epsilon m} \sum_{k=1}^m \max\left\{0, -x_k - \alpha\right\} \right\}, \tag{22}$$

which amounts to averaging the tails of $-\mathbf{x} = -(x_1, \ldots, x_m)$, i.e., entries that are larger than the $(1 - \epsilon)$ quantile, a.k.a. value-at-risk (VaR). In particular, for $\epsilon \leq \tfrac{1}{m}$, CVaR coincides with MGDA, whereas for $\epsilon \geq 1 - \tfrac{1}{m}$, CVaR reduces to Linear Scalarization. Other values of $\epsilon$ provide different interpolations between these two extreme cases. With an appropriate choice of $\epsilon$, we can control the influence of the extreme values in $\mathbf{x}$. It is easy to derive the Fenchel conjugate

$$s^*(-\boldsymbol{\lambda}) = \mathrm{CVaR}_\epsilon^*(-\boldsymbol{\lambda}) = \begin{cases} 0, & \text{if } \boldsymbol{\lambda} \in \Delta \text{ and } \boldsymbol{\lambda} \leq C \\ \infty, & \text{otherwise} \end{cases}, \tag{23}$$

---

[2]We thank a reviewer for bringing these relevant works to our attention; see Appendix A for more discussion of the two methods and how to apply our framework to them.

where $C := \frac{1}{\epsilon m}$, which is similar to that of MGDA: the only difference is the cap constraint $\boldsymbol{\lambda} \leq C$, which limits the contribution of each component gradient. Thus, the implementation of CVaR (which we refer to as *Capped MGDA*) closely mirrors that of MGDA, with the additional cap constraint imposed when solving the MGDA dual quadratic program; see Appendix A.2.1 for a detailed derivation of the dual. To our knowledge, CVaR, or more generally, convex-risk-measure-based directions have not been explored before in MOO.

**Convergence of convex-risk-measure-based directions.** With $r(\rho) = \frac{1}{2}\rho^2$, we have $\beta_t \equiv 1$ in Theorem 4. Since $\boldsymbol{\lambda} \in \Delta$, we have $c_t = \frac{1}{2}$ in Theorem 4. Applying Corollary 1, we immediately obtain the $O(1/\sqrt{t})$ rate of convergence to Pareto stationarity for all such directions.

## 5 EXPERIMENTS

We conduct experiments on both synthetic problems and realistic benchmarks to study existing non-conflicting gradient aggregators, both individually and under mixed aggregator scheduling (MAS), as well as the newly proposed *Capped MGDA*. These experiments examine convergence behavior or robustness under adversarial conditions, serving to validate our theoretical findings rather than to rank methods[3]. Further details and discussions are provided in Appendix C.

### 5.1 NON-CONFLICTING GRADIENT AGGREGATORS

For non-conflicting gradient aggregators, we conduct experiments on synthetic problems (VLMOP2 and Omnitest) and on a realistic fairness classification benchmark. In line with our theoretical findings (Theorem 2), we examine convergence using the measure of Pareto stationarity $\gamma(\mathbf{w})$.

**Methods.** We evaluate four non-conflicting aggregation schemes (Quinton & Rey, 2024): MGDA, DualProj, UPGrad, and Nash-MTL. For each, except MGDA (whose update direction already lies in the convex hull), we also include a normalized variant (denoted with a star) where $\mathbf{d}$ is rescaled to lie in the convex hull of gradients. In addition, we consider *Mixed Aggregator Scheduling (MAS)*, which alternates among non-conflicting methods according to a prescribed schedule (see Algorithm 1, Appendix A). Different non-conflicting aggregators have complementary properties: for instance, MGDA may stall at suboptimal Pareto-stationary points (Hu & Yu 2025), whereas UPGrad can continue making progress; Nash-MTL, though more expensive, provides scale-invariant conflict resolution. Simple schedules such as uniform random selection at each iteration (`Rand`), or round-robin every $n$ iterations (`RR(n)`) serve as natural baselines for mixing these methods.

**Synthetic problems setup.** We evaluate on two common synthetic MOO benchmarks: VLMOP2 (van Veldhuizen & Lamont, 1999) and Omnitest (Deb & Tiwari, 2008), each with five random seeds.

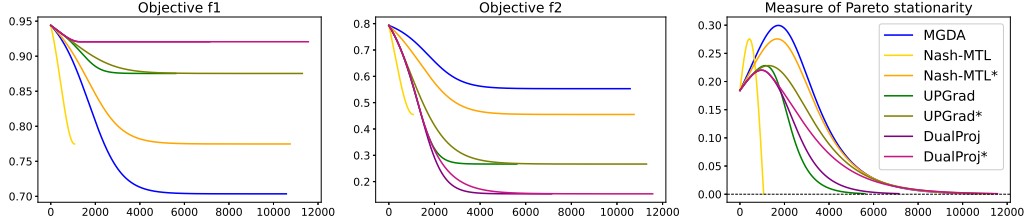

Figure 2: Dynamics of non-conflicting aggregators on VLMOP2. **Left**: objective $f_1$; **Middle**: objective $f_2$; **Right**: Pareto stationarity measure $\gamma(\mathbf{w}_t)$.

**Results.** Figure 2 reports the optimization dynamics of four non-conflicting aggregation methods on VLMOP2, using both the original directions and their normalized counterparts (denoted by '*'). Since normalization to the convex hull removes scale differences and places all methods on a comparable footing, these normalized variants are the ones most indicative of their intrinsic convergence behavior, and indeed they exhibit similar asymptotic rates. The unnormalized variants appear to converge faster, but only because their larger $\|\mathbf{d}_t\|$ means effectively using a larger step size. This is especially visible for Nash-MTL, whose update norm stays constant even when close to stationarity.

---

[3]Nonconvex MOO usually yields incomparable Pareto stationary solutions, and even in convex settings, Pareto optimal solutions are generally not directly comparable.

Table 2: Fairness classification on Adult dataset. Results averaged over 4 random seeds.

|  | Accuracy (%)↑ | DEO1 (%)↓ | DEO2 (%)↓ |
| --- | --- | --- | --- |
| MGDA | $75.70 \pm 0.06$ | $61.04 \pm 0.59$ | $7.56 \pm 0.28$ |
| Nash-MTL* | $76.19 \pm 0.11$ | $58.96 \pm 0.96$ | $7.08 \pm 0.41$ |
| DualProj | $\mathbf{79.02 \pm 0.11}$ | $53.97 \pm 0.37$ | $\mathbf{6.20 \pm 0.06}$ |
| DualProj* | $78.55 \pm 0.05$ | $\mathbf{53.69 \pm 0.30}$ | $6.63 \pm 0.08$ |
| UPGrad | $77.66 \pm 0.11$ | $62.36 \pm 0.58$ | $6.97 \pm 0.06$ |
| UPGrad* | $76.75 \pm 0.08$ | $60.18 \pm 0.70$ | $7.37 \pm 0.06$ |
| MAS-Rand | $76.42 \pm 0.07$ | $60.90 \pm 0.48$ | $7.45 \pm 0.10$ |
| MAS-RR(1k) | $76.41 \pm 0.09$ | $60.94 \pm 0.57$ | $7.48 \pm 0.12$ |
| MAS-RR(2.5k) | $76.25 \pm 0.07$ | $60.56 \pm 0.63$ | $7.69 \pm 0.12$ |

In all cases, the Pareto-stationarity measure $\gamma(\mathbf{w}_t)$ (see (6)) consistently converges to zero, aligning with the guarantee in Theorem 2. Figure 3 further demonstrates the dynamics of our mixed-aggregator scheduling (MAS) scheme using random and round-robin schedules. The trajectories confirm that switching among different non-conflicting methods within a single optimization run still ensures convergence to Pareto-stationarity, with end-phase asymptotics comparable to individual methods. This provides empirical support for the validity of MAS as suggested by our theory.

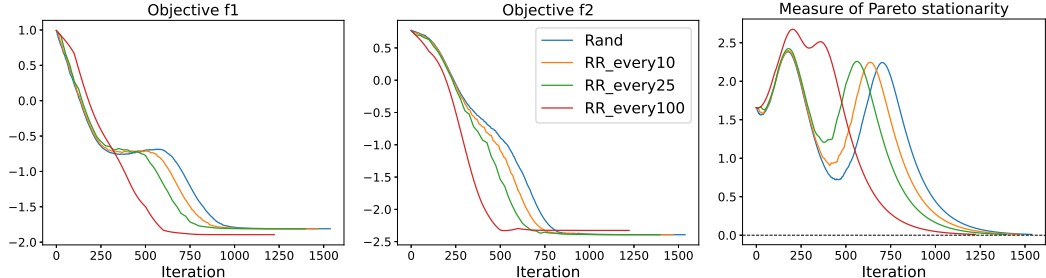

Figure 3: Dynamics of mixed aggregator scheduling on Omnitest. **Left**: objective $f_1$; **Middle**: objective $f_2$; **Right**: Pareto stationarity measure $\gamma(\mathbf{w}_t)$.

**Fairness classification setup.** We follow LibMOON (Zhang et al., 2024) for evaluating fairness classification on the Adult dataset, where a 4-layer MLP is trained to predict the income level. The objectives are binary cross-entropy (utility) and smoothed relaxations of Difference of Equalized Odds[4] (fairness) with $Y = 1$ (DEO1) and $Y = 0$ (DEO2). Details are provided in Appendix C.

**Results.** Table 2 reports the fairness results for the aforementioned methods[5]. MAS achieves intermediate performance: it outperforms some methods in fairness and accuracy, but falls short of others. This positions MAS a natural baseline—its use of all aggregators within a single trial provides a representative "average" level of performance against which other methods can be compared.

## 5.2 CAPPED MGDA

**Setup.** We evaluate capped MGDA in a federated learning setting on the CIFAR-10 dataset, with $m = 10$ clients. Each client holds distinct non-i.i.d. data partitions, and the objectives $f_i$ correspond to their individual prediction utilities. We consider an adversarial scenario where, during gradient aggregation, a malicious attacker contributes a flipped gradient (with some noise), opposite to one client's direction. The goal is to compare the robustness and effectiveness of capped MGDA against MGDA in the presence of such adversarial gradients. Further details are provided in Appendix C.

**Results.** As shown in Figure 4 (Top Left), capped MGDA achieves substantially higher per-client test accuracies than MGDA in the adversarial FL setting. This highlights MGDA's vulnerability to adversarial gradients. Top Right panel further confirms that this gap is specific to the adversarial

---

[4]Equalized odds with $Y = 1$ (used in DEO1) is also referred to as equal opportunity (Hardt et al., 2016).

[5]Nash-MTL is excluded due to instability; see the discussion in the Appendix C.

scenario: in the standard (no-attack) FL setting, MGDA and capped MGDA perform comparably, indicating that capping does not degrade performance when no adversary is present.

MGDA's vulnerability arises from its min-norm update: when opposite gradients are present, MGDA assigns large weights (near $\frac{1}{2}$) to them, resulting in a much smaller update direction $\mathbf{d}$, as seen in Figure 4 (Bottom Left, orange curve). This leads to ineffective progress under adversarial attacks. By limiting each gradient's maximum contribution, capped MGDA avoids these extreme allocations and yields larger, more meaningful update directions. Finally, Figure 4 (Bottom Right) shows that capped MGDA is, in general, *not* a non-conflicting aggregator: smaller values of $C$ lead to the curves shifting further below zero, indicating more severe gradient conflicts.

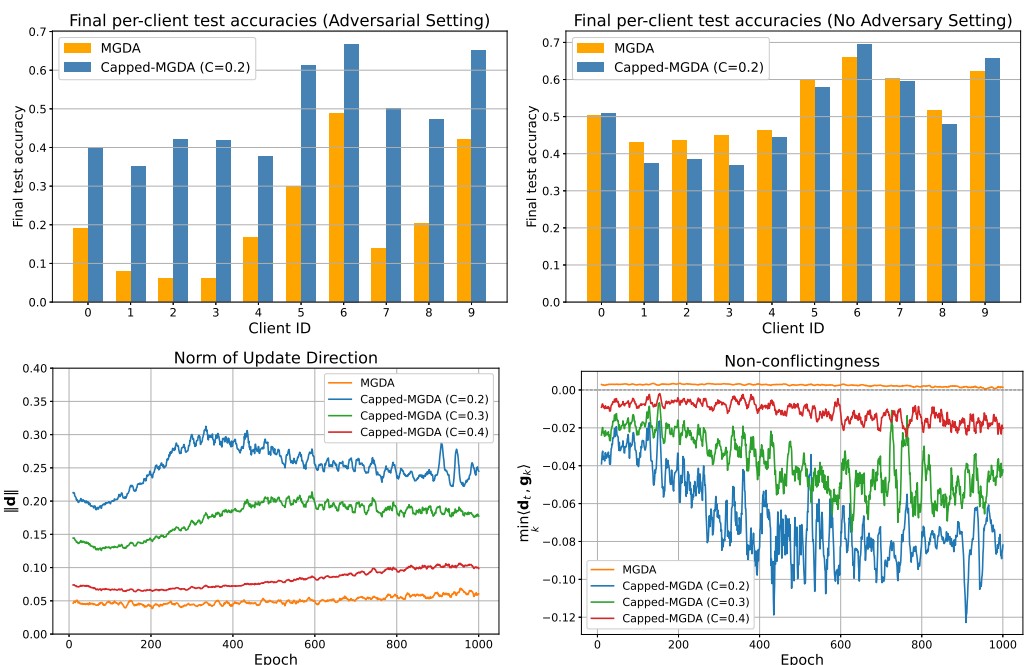

Figure 4: Capped MGDA vs. MGDA in adversarial federated learning on CIFAR-10 (1000 epochs). **Top**: Per-client test accuracies (clients 0–9) for MGDA and Capped-MGDA under both adversarial and standard FL settings. **Bottom Left**: Norm of the update direction $\mathbf{d}_t$ throughout adversarial FL training. **Bottom Right**: Non-conflictingness of the update direction, measured by $\min_k \langle \mathbf{d}_t, \mathbf{g}_k \rangle$ (positive values indicate non-conflicting updates), during adversarial FL training.

## 6 CONCLUSION

We present a unifying framework for gradient aggregation in multi-objective optimization. Central to our framework is an alignment condition that simplifies convergence analysis in MOO, leading to theorems with simpler and more intuitive conditions and, to our knowledge, the first comprehensive answer to the question of which directions guarantee convergence in MOO. We further summarize and clarify a wide range of existing aggregation methods, showing that ensuring a non-conflicting direction is sufficient for convergence. Inspired by Theorem 2, we also propose the Mixed Aggregator Scheduling (MAS) strategy and demonstrate that mixing different aggregators within a single training run yields a valid and practically algorithm. In Theorem 4, we introduce a subproblem-based construction of convergent directions, which not only covers many existing methods but also enables the design of new ones, in particular, capped MGDA.

Limitations and future work include tightening rates under stronger curvature (e.g., PL/strong convexity), extending the analysis to stochastic and constrained/non-smooth settings, and exploring mixed aggregator scheduling more comprehensively.

ETHICS STATEMENT

We use ChatGPT only for polishing words and sentences. We adhere to the ICLR Code of Ethics.

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

---

**Algorithm 1:** Multi-Objective Descent Algorithm with Mixed Aggregator Scheduling

---

**Input:** Objectives $\mathbf{f}$, pool of aggregation schemes $\{\mathcal{A}_i\}$, initializer $\mathbf{w}_0$, learning rate $\eta_t$.

1 **for** $t = 0, 1, \ldots, T-1$ **do**
2     $J_{\mathbf{f}}(\mathbf{w}_t) \leftarrow [\nabla f_1(\mathbf{w}_t), \ldots, \nabla f_m(\mathbf{w}_t)]$              // compute gradients
3     Choose $\mathcal{A}_j$ from the pool   // select an aggregator based on a schedule
4     $\mathbf{d}_t \leftarrow \mathcal{A}_j(J_{\mathbf{f}}(\mathbf{w}_t))$               // aggregate gradients using $\mathcal{A}_j$
5     $\mathbf{w}_{t+1} \leftarrow \mathbf{w}_t - \eta_t \cdot \mathbf{d}_t$                      // update

**Output:** final weights $\mathbf{w}_T$

---

## A   More on gradient aggregations

### A.1   Existing aggregators

Here, we review commonly used multi-objective gradient aggregation methods from the literature and present their optimization subproblem formulations under (12) (with the corresponding choices of $s$ and $r$), along with their algorithmic formulations as implemented in practice. We also provide discussions and insights on methods that do not neatly fit into our framework (e.g., PCGrad Yu et al. 2020 and IMTL-G Liu et al. 2021b).

#### A.1.1   (Uniform) Linear scalarization

The primal optimization subproblem formulation is

$$\operatorname*{argmin}_{\mathbf{d}} \; -\frac{1}{m}\sum_{k=1}^{m}\langle \mathbf{d}, \mathbf{g}_k\rangle + \tfrac{1}{2}\|\mathbf{d}\|^2, \tag{24}$$

which corresponds to choosing $s(\mathbf{x}) = -\frac{1}{m}\mathbf{1}^\top \mathbf{x}$ and $r(\|\cdot\|) = \frac{1}{2}\|\cdot\|^2$ in (12).

The resulting gradient aggregation rule (used in practice) is

$$\mathbf{d} = \frac{1}{m}\sum_{k=1}^{m}\mathbf{g}_k. \tag{25}$$

Linear scalarization is not necessarily a non-conflicting aggregator (it is easy to construct cases where the averaged gradient $\mathbf{d}$ has a negative inner product with some component gradient). Nevertheless, its convergence to Pareto stationarity follows directly by taking $F = \frac{1}{m}\sum_i f_i$ and $c_t = 1$ in Theorem 1. Moreover, the scalarization weights need not be fixed at $\frac{1}{m}$; any conic scalarization preserves this convergence property, and it is common to tune the weights for better performance.

**Convergence guarantees of linear scalarization.** The aggregation rule in (25) corresponds precisely to performing gradient descent on the scalarized objective $F = \sum_i f_i$. Consequently, its convergence behavior follows that of standard gradient descent in *smooth* single-objective optimization, with a rate of $O(1/\sqrt{t})$ for non-convex objectives and $O(1/t)$ for convex objectives. Our framework, when applicable[6], recovers these same (optimal) rates, as established in the main paper.

#### A.1.2   Multiple Gradient Descent Algorithm (MGDA)

The primal optimization subproblem formulation of MGDA (Mukai, 1980; Fliege & Svaiter, 2000; Désidéri, 2012) is

$$\operatorname*{argmin}_{\mathbf{d}} \; \max_{k} -\langle \mathbf{d}, \mathbf{g}_k\rangle \tag{26}$$

which corresponds to choosing $s(\mathbf{x}) = \max_k(-x_k)$, $r(\|\cdot\|) = \frac{1}{2}\|\cdot\|^2$ in (12).

---

[6]In the non-convex case, Theorem 1 applies directly; however, in the convex case, gradient descent is not necessarily non-conflicting, and thus Theorem 3 cannot be invoked directly.

The resulting gradient aggregation rule (used in practice) is

$$\mathbf{d} = J\boldsymbol{\lambda}_*, \quad \text{where } \boldsymbol{\lambda}_* = \underset{\boldsymbol{\lambda} \in \Delta}{\operatorname{argmin}} \|J^\top \boldsymbol{\lambda}\|^2. \tag{27}$$

MGDA is automatically a non-conflicting aggregator, which is easy to see from the primal perspective e.g., (Fliege & Svaiter, 2000). Its update $\mathbf{d}_t$ is also in the convex full of gradients by definition of the dual problem. Thus, its convergence is immediate from Theorem 2.

**Convergence guarantees of MGDA.** To the best of our knowledge, the most complete complexity analysis of MGDA to date is provided by Fliege et al. (2019). Our framework extends this analysis to a broader class of gradient aggregation schemes. When specialized to MGDA, our general results (Theorems 2 and 3) apply without requiring additional assumptions and recover the same convergence rates as those established by Fliege et al. (2019) for both non-convex and convex settings.

### A.1.3 NASH BARGAINING MULTI-TASK LEARNING (NASH-MTL)

The primal optimization subproblem formulation of Nash-MTL (Navon et al., 2022) is

$$\underset{\|\mathbf{d}\| \leq \epsilon}{\operatorname{argmin}} -\sum_k \log \langle \mathbf{d}, \mathbf{g}_k \rangle, \tag{28}$$

which corresponds to choosing $s(\mathbf{x}) = -\sum_k \log x_k$, $r(\|\cdot\|) = \iota_{\mathcal{B}_\epsilon}(\cdot) := \begin{cases} 0, & \|\mathbf{d}\| \leq \epsilon \\ +\infty, & \text{otherwise} \end{cases}$

Note that this optimization formulation is equivalent to the $-(\prod_k x_k)^{1/m}$ we presented in the main paper, by moving the negative sign out and take the log. Also, the hard ball constraint $\|\mathbf{d}\| \leq \epsilon$ in (28) can be equivalently replaced with regularization $\frac{1}{2}\|\mathbf{d}\|^2$ which will result in the same aggregation rule as in (Eq), up to constant scaling.

The resulting gradient aggregation rule (used in practice) is

$$\mathbf{d} = J\boldsymbol{\lambda}_*, \quad \text{where } J^\top J \boldsymbol{\lambda}_* = \frac{\mathbf{1}}{\boldsymbol{\lambda}_*} \tag{Eq}$$

Nash-MTL is a non-conflicting aggregator, which can be seen directly from the primal perspective. In the official implementation, $\mathbf{d}$ is clipped to satisfy $\|\mathbf{d}\| = \epsilon$ for some fixed $\epsilon$ (default 1). Our experiments show that applying convex-hull regularization (the variant denoted Nash-MTL*), which rescales $\boldsymbol{\lambda}_*$ to lie in $\Delta$, greatly stabilizes and smooths training. Thus, convex-hull regularization not only ensures convergence via Theorem 2 but is also empirically preferable. Moreover, our convergence-analysis framework removes the need for the linearly independent gradients assumption required in the original work.

**Convergence guarantees of Nash-MTL.** For the *non-convex* case, Navon et al. (2022) established convergence under three main assumptions: (1) linear independence of the gradients $\{\mathbf{g}_k^{(t)}\}$ at each iterate $\mathbf{w}^{(t)}$ and at the limit, (2) Lipschitz smooth and lower-bounded objectives $f_k$, and (3) bounded sub-level sets. Under these conditions, they showed that the sequence $\{\mathbf{w}^{(t)}\}_{t=1}^\infty$ admits a subsequence converging to a Pareto-stationary point. In contrast, under our framework (e.g., Theorem 2), assumptions (1) and (3) are no longer required. Instead, we establish an explicit convergence rate of $O(1/\sqrt{t})$ with respect to the degree of Pareto stationarity $\gamma(\mathbf{w}_t)$, merely from assumption 2.

For the *convex* case, both Navon et al. (2022) and our analysis (Theorem 3) rely on the same standard assumptions. While Navon et al. (2022) proves convergence of $\mathbf{w}_t$ to a weakly Pareto-optimal solution, our framework provides a rate of $O(1/t)$ in terms of the function value gap to optimality.

### A.1.4 FAIR RESOURCE ALLOCATION IN MTL (FAIRGRAD)

To provide an $\alpha$-fair framework for MTL (and MOO in general), FairGrad (Ban & Ji, 2024a) proposes the following subproblem to optimize:

$$\underset{\|\mathbf{d}\| \leq \epsilon}{\operatorname{argmin}} -\sum_k \frac{(\langle \mathbf{d}, \mathbf{g}_k \rangle)^{1-\alpha}}{1-\alpha}, \text{ s.t. } \langle \mathbf{d}, \mathbf{g}_k \rangle \geq 0, \ \forall k. \tag{29}$$

which has the resulting gradient aggregation rule:

$$\mathbf{d} = J\boldsymbol{\lambda}_*, \quad \text{where } J^\top J\boldsymbol{\lambda}_* = \boldsymbol{\lambda}_*^{-1/\alpha} \tag{30}$$

Similar to our power-mean-based formulation, FairGrad can recover Linear Scalarization, Nash-MTL, harmonic mean ($p = -1$), and MGDA when $\alpha \to 0, 1, 2, \infty$. Note that $-\frac{x^{1-\alpha}}{1-\alpha}$ is decreasing and convex for $x > 0$ and $\alpha \geq 0$, so Theorem 4 applies as long as the constraint $\mathbf{d} \in B_\epsilon$ is replaced by a quadratic penalty term $\frac{1}{2}\|\mathbf{d}\|^2$, though its implementation may be cumbersome in practice.

**Convergence guarantees of FairGrad.** The original FairGrad work provides a convergence analysis in the non-convex setting. Similar to the proof of Nash-MTL, it requires the following assumptions: (1) linear independence of the gradients at each iterate, (2) Lipschitz smooth and lower-bounded objectives $f_k$, and (3) bounded sub-level sets. Under these assumptions, the authors establish subsequence convergence.

Our framework (i.e. Theorem 2) applies because FairGrad produces non-conflicting directions, and the convex-hull requirement can be satisfied by rescaling $\mathbf{d}_t$ and absorbing the resulting constant into the step size $\eta_t$, we note that this effectively introduces a dynamic step size. The original FairGrad proof also accommodates a dynamic step size, which further justifies our modification. Under our framework, assumptions (1) and (3) are no longer required. Relying only on assumption (2), we can obtain an explicit convergence rate of $O(1/\sqrt{t})$ with respect to the degree of Pareto stationarity $\gamma(\mathbf{w}_t)$, albeit without guaranteeing pointwise subsequence convergence of $\mathbf{w}_t$.

### A.1.5 PERFORMANCE-INFORMED VARIANCE REDUCTION GRADIENT AGGREGATION (PIVRG)

PIVRG (Qin et al., 2025) proposes to minimize the (weighted) mean of the inverse utilities as the subproblem:

$$\underset{\|\mathbf{d}\| \leq \epsilon}{\text{argmin}} \frac{1}{m} \sum_{k=1}^{m} \frac{\omega_k}{\langle \mathbf{d}, \mathbf{g}_k \rangle} \tag{31}$$

where $\omega_k$ are dynamic coefficients incorporating performance-level information.

For gradient aggregation rule used in practice, they solve for:

$$\mathbf{d} = J\boldsymbol{\lambda}_*, \quad \text{where } J^\top J\boldsymbol{\lambda}_* = \left(\frac{\boldsymbol{\omega}}{\boldsymbol{\lambda}_*}\right)^{\frac{1}{2}} \tag{32}$$

We note that PIVRG is conceptually very close to FairGrad (with $\alpha = 2$), though PIVRG introduces a novel design of the weighting coefficients $\boldsymbol{\omega}$ to achieve improved variance reduction.

**Convergence guarantees of PIVRG.** The theoretical assumptions and convergence results of PIVRG largely mirror those of FairGrad, and our general framework applies in a similar fashion. To avoid redundancy, we refer the reader to the FairGrad section for a detailed discussion and comparison of the convergence guarantees.

### A.1.6 UNCONFLICTING PROJECTION OF GRADIENTS (UPGRAD)

The primal optimization subproblem formulation of UPGrad (Quinton & Rey, 2024) is

$$\underset{\mathbf{d}}{\text{argmin}} -\frac{1}{m} \sum_{k=1}^{m} \langle \mathbf{d}, \mathbf{p}_k \rangle + \frac{1}{2}\|\mathbf{d}\|^2, \tag{33}$$

$$\text{where} \quad \mathbf{p}_k := \text{P}_{\text{cone}^*(J)}(\mathbf{g}_k), \ J\boldsymbol{\alpha}_k := \mathbf{p}_k. \tag{34}$$

which corresponds to choosing $s(\mathbf{x}) = -(\frac{1}{m}\sum_k \boldsymbol{\alpha}_k)^\top \mathbf{x}, \ r(\|\cdot\|) = \frac{1}{2}\|\cdot\|^2$ in (12).

The resulting gradient aggregation rule (used in practice) is

$$\mathbf{d} = \frac{1}{m} \sum_k \mathbf{p}_k = J\left(\frac{1}{m} \sum_{k=1}^{m} \boldsymbol{\alpha}_k\right). \tag{35}$$

which first projects each gradient onto the dual cone $\{\mathbf{d} : J^\top \mathbf{d} \geq \mathbf{0}\}$ and then averages them.

(I) UPGrad is a non-conflicting aggregator since it first projects all gradients onto the dual cone.

(II) UPGrad is closely related to PCGrad (Yu et al., 2020), and in fact coincides with it when $m = 2$. We discuss this connection in detail in the PCGrad section.

**Convergence guarantees of UPGrad.** For the *non-convex* setting, Quinton & Rey (2024) (Appendix B.4) established an $O(1/\sqrt{t})$ convergence rate under the same assumptions of Lipschitz smoothness and lower boundedness of each $f_i$ as those required by our framework (e.g., Corollary 2). Although the rate is identical, their analysis relies on a method-specific proof, whereas our framework provides a more general and conceptually unified derivation.

For the *convex* setting, both Quinton & Rey (2024) and our analysis establish an $O(1/t)$ convergence rate in terms of the function value gap, under the same assumptions of Lipschitz smoothness and convexity. While rate is the same, our upper bound is slightly tighter. Additionally, under the extra assumptions of (1) a bounded Pareto front and (2) bounded coefficients $\boldsymbol{\lambda}_t$, Quinton & Rey (2024) used a method-specific proof to show the convergence of $\mathbf{f}(\mathbf{w}_t)$ to $\mathbf{f}^*$. This result can also be obtained within our framework, through a direct application of Theorem 3 and upper bounding $\boldsymbol{\lambda}$.

### A.1.7 DUALPROJ

The primal optimization subproblem formulation of DualProj (Lopez-Paz & Ranzato, 2017) is

$$\underset{\mathbf{d}}{\arg\min} - \sum_{k=1}^{m} \alpha_k \langle \mathbf{d}, \mathbf{g}_k \rangle + \frac{1}{2}\|\mathbf{d}\|^2, \tag{36}$$

$$\text{where} \quad J\boldsymbol{\alpha} := \mathrm{P}_{\mathrm{cone}^*(J)}\left(\frac{1}{m}\sum_k \mathbf{g}_k\right). \tag{37}$$

The resulting gradient aggregation rule (used in practice) is

$$\mathbf{d} = J\boldsymbol{\alpha} \tag{38}$$

DualProj is a non-conflicting aggregator since it projects a convex combination of gradients onto the dual cone, see Proposition 1.

**Convergence guarantees of DualProj.** The original work of Lopez-Paz & Ranzato (2017) does not appear to provide a formal convergence analysis. Within our framework, Corollary 2 establishes an $O(1/\sqrt{t})$ convergence rate for DualProj in terms of the Pareto stationarity measure $\gamma(\mathbf{w}_t)$ for the non-convex setting. For the convex setting, we can apply Theorem 3 and establish a $O(1/t)$ rate.

### A.1.8 CONFLICT-AVERSE GRADIENT DESCENT (CAGRAD)

The primal optimization subproblem formulation of CAGrad (Liu et al., 2021a) is

$$\underset{\mathbf{d}}{\arg\min} \ \underset{k}{\max} - \langle \mathbf{d}, \mathbf{g}_k \rangle, \quad \text{s.t.} \ \|\mathbf{d} - \mathbf{g}_0\| \leq c\|\mathbf{g}_0\|. \tag{39}$$

where $\mathbf{g}_0 := \frac{1}{m}\sum_k \mathbf{g}_k$, and $c$ is a pre-specified hyper-parameter that controls the radius of the ball constraint centered around the average gradient $\mathbf{g}_0$.

The resulting gradient aggregation rule (used in practice) is

$$\mathbf{d} = \mathbf{g}_0 + \epsilon\, \mathbf{g}_{\boldsymbol{\lambda}_*} \tag{40}$$

where $\epsilon := c\|\mathbf{g}_0\|$, $\mathbf{g}_{\boldsymbol{\lambda}_*} := J\boldsymbol{\lambda}_*$, and $\boldsymbol{\lambda}_*$ is the solution to

$$\underset{\boldsymbol{\lambda}\in\Delta}{\arg\min} \ \langle \mathbf{g}_0, J\boldsymbol{\lambda} \rangle + \epsilon\|J\boldsymbol{\lambda}\|. \tag{41}$$

CAGrad is a direction-oriented variant of MGDA. Due to the existence of the ball constraint, it is no longer a non-conflicting aggregator when $c < 1$; however, it is non-conflicting when $c \geq 1$ (since $\mathbf{d} = \mathbf{0}$ is a feasible point).

**Convergence guarantees of CAGrad.** The convergence behavior of CAGrad depends on the choice of the hyper-parameter $c$. The original work of Liu et al. (2021a) considers the non-convex setting, and:

(1) For $c \geq 1$, the authors only showed that the fixed point of CAGrad is Pareto stationary. In contrast, by noting that CAGrad is *non-conflicting* in this case, our framework (Theorem 2) can be directly applied to *strengthen* the guarantee to an $O(1/\sqrt{t})$ convergence rate measured by $\gamma(\mathbf{w}_t)$.

(2) For $0 \leq c < 1$, Liu et al. (2021a) established an $O(1/\sqrt{t})$ rate to stationarity of the averaged objective $\frac{1}{m} \sum_{i=1}^{m} f_i$, and thus to Pareto stationarity of $\mathbf{f}$. We point out this convergence guarantee is a matter of fact of the ball constraint rather than the objective, and that we can directly apply the angle constraint (10) in Theorem 1 (with $F = \frac{1}{m} \sum_{i=1}^{m} f_i$) to obtain the same result.

In the convex setting, the original work does not provide extra convergence guarantees. However, when $c \geq 1$, CAGrad remains non-conflicting, and therefore our framework (Theorem 3) can be applied to this case, and establish a rate of $O(1/t)$.

### A.1.9  PROJECTING CONFLICTING GRADIENTS (PCGRAD)

PCGrad (Yu et al., 2020) aims to fix each gradient $\mathbf{g}_k$ by initializing $\mathbf{g}_k^{\mathrm{PC}} \leftarrow \mathbf{g}_k$, and then iteratively projecting it onto the normal planes of the other gradients, for one pass over $[m]$ only:

$$\text{for } i \in [m], \ \mathbf{g}_k^{\mathrm{PC}} \leftarrow \mathbf{g}_k^{\mathrm{PC}} + \frac{(-\mathbf{g}_k^{\mathrm{PC}} \cdot \mathbf{g}_i)_+}{\|\mathbf{g}_i\|^2} \mathbf{g}_i \tag{42}$$

and finally gather and average the $\mathbf{g}_k^{\mathrm{PC}}$:

$$\mathbf{d} = \frac{1}{m} \sum_{k=1}^{m} \mathbf{g}_k^{\mathrm{PC}}. \tag{43}$$

We highlight several important points:

- When the number of objectives is $m = 2$, PCGrad is equivalent to UPGrad, since in this case projection to the 'dual cone' coincides with projection to the 'normal plane of the other gradient'. Notably, $m = 2$ is also the only setting in which the PCGrad paper provides theoretical guarantees. Thus, for two-objective optimization, PCGrad can be regarded as UPGrad, the latter being studied more extensively in this paper.

- When $m \geq 3$, there is no guarantee that the resulting $\mathbf{g}_k^{\mathrm{PC}}$ (and thus $\mathbf{d}$) is a non-conflicting direction, because PCGrad performs only a single pass of iterative projections rather than continuing until $\mathbf{g}_k^{\mathrm{PC}}$ lies in the dual cone. For a counter-example, let

$$\mathbf{g}_1 = (1, \sqrt{3}, z), \quad \mathbf{g}_2 = (-2, 0, z), \quad \mathbf{g}_3 = (1, -\sqrt{3}, z), \tag{44}$$

  where $z$ is a small positive constant (e.g., $z = 0.1$). Applying the standard PCGrad procedure in order and averaging the adjusted gradients, the resulting aggregated direction $\mathbf{d}$ has a negative inner product with $\mathbf{g}_1$.

- PCGrad can be modified to repeatedly project until $\mathbf{g}_k^{\mathrm{PC}}$ lies in the dual cone, and we name this new variant *PCGrad+* (see Algorithm 2). In this case, PCGrad+ again resembles UPGrad, except that UPGrad performs a one-step projection directly onto the dual cone $C^*$, whereas PCGrad+ repeatedly performs alternating projections onto the half-spaces $H_i = \{\mathbf{z} : \langle \mathbf{g}_i, \mathbf{z} \rangle \geq 0\}$, whose intersection is the dual cone, i.e. $C^* = \bigcap_{i=1} H_i$. PCGrad+ is also sensitive to the ordering of objectives, which is arguably an undesirable property.

---

**Algorithm 2:** PCGrad$^+$ Aggregation

---

**Input:** Gradients $\mathbf{g}_k := \nabla f_k(\mathbf{w})$ of each objective $f_k$.

**1 for** $k \in [m]$ **do**

  **2**    $\mathbf{g}_k^{\mathrm{PC}} \leftarrow \mathbf{g}_k$                            `// Initialize projected gradient`

  **3**    **repeat**

  **4**      `// key distinction from PCGrad`

  **5**      **for** $i \in [m]$ **do**

  **6**        $\mathbf{g}_k^{\mathrm{PC}} \leftarrow \mathbf{g}_k^{\mathrm{PC}} + \dfrac{(-\mathbf{g}_k^{\mathrm{PC}} \cdot \mathbf{g}_i)_+}{\|\mathbf{g}_i\|^2}\mathbf{g}_i$          `// Resolve conflicts via`

            `projection`

  **7**   **until** $\mathbf{g}_k^{\mathrm{PC}}$ *is non-conflicting with all* $\mathbf{g}_i$

**Output:** Aggregated direction $\mathbf{d} \leftarrow \frac{1}{m}\sum_{k=1}^m \mathbf{g}_k^{\mathrm{PC}}$.

---

**Convergence guarantees of PCGrad(+).** To the best of our knowledge, Yu et al. (2020) provided a convergence analysis of PCGrad under reasonable assumptions *only* for the case of two objectives ($m = 2$). As noted above, when $m = 2$, PCGrad coincides with UPGrad, and therefore our framework for UPGrad applies directly to PCGrad in this special case.

When $m \geq 3$, however, the output of PCGrad aggregation (1) depends on the order of the input objectives and (2) is not guaranteed to be non-conflicting, both of which pose challenges for a general convergence analysis (if attainable at all).

To address this, we introduce a modified variant, termed *PCGrad+*, which repeatedly performs the gradient-fixing projections until the resulting direction $\mathbf{d}_t$ lies within the dual cone. For PCGrad+, our general results, Theorem 2 (non-convex) and Theorem 3 (convex), can be readily applied to establish convergence for arbitrary $m \geq 3$.

### A.1.10   Impartial Multi-task Learning Gradient (IMTL-G)

Let $\mathbf{n} = [\|\mathbf{g}_1\|, \ldots, \|\mathbf{g}_m\|]^\top$. IMTL-G (Liu et al., 2021b) aggregates the gradients as

$$\mathbf{d} = J\boldsymbol{\lambda}, \quad \text{where} \quad \boldsymbol{\lambda} = (J^\top J)^\dagger \mathbf{n}, \tag{45}$$

and then applies a (possibly negative) rescaling to enforce $\sum_i \lambda_i = 1$ to get normalized update $\tilde{\mathbf{d}}$.

Interestingly, a recent work (Liu et al., 2025) on Physics-Informed Neural Networks is closely related to IMTL-G. However, it does not enforce the constraint $\sum_i \lambda_i = 1$, and instead introduces a different form of normalization.

It's worth noting that:

(I) The update direction $\mathbf{d}$ is *not necessarily* non-conflicting unless the gradients are normalized. As a counterexample, consider the following Jacobian $J$ (whose row vectors are the gradients), where $\mathbf{d}$ turns out to be conflicting with all gradients:

$$J = \begin{pmatrix} 8.660254 & -5 & 0.0 \\ -8.660254 & -5 & 0.0 \\ 0.0 & -0.99498744 & 0.1 \end{pmatrix} \tag{46}$$

(II) $\mathbf{d}$ is *not necessarily* in the cone either. As a counterexample, consider the same matrix $J$, but with the first two rows divided by $10$.

**Convergence guarantees of IMTL-G.** The original work of Liu et al. (2021b) does not appear to include a formal convergence analysis. Due to the irregularities and drawbacks discussed above, IMTL-G is also difficult to incorporate into our theoretical framework, since its update direction $\mathbf{d}$ does not necessarily lie within either the cone or the dual cone.

### A.1.11 RANDOM GRADIENT WEIGHTING (RGW)

Random Gradient Weighting (Lin et al., 2022) uses random weighting to aggregate the gradients:

$$\mathbf{d} = J \cdot \text{softmax}(\boldsymbol{\xi}), \text{ where } \boldsymbol{\xi} \sim \mathcal{N}(\mathbf{0}, I) \tag{47}$$

We can formulate a primal optimization subproblem:

$$\underset{\mathbf{d}}{\text{argmin}} - \sum_{k=1}^{m} \xi_k \langle \mathbf{d}, \mathbf{g}_k \rangle + \frac{1}{2} \|\mathbf{d}\|^2. \tag{48}$$

**Convergence guarantees of RGW.** Unlike all aforementioned methods, RGW admits no fixed point (except for the degenerate case where all $\mathbf{g}_k = \mathbf{0}$), and therefore cannot terminate or converge at any non-trivial Pareto stationary point. This poses an inherent challenge to establishing convergence guarantees for RGW. The original work provides only an upper bound on the function value gap, which can be unbounded unless all gradients are assumed to be bounded. Moreover, the bound does not vanish as $t \to \infty$. In summary, we find it difficult to establish a theoretical convergence guarantee for this method without imposing very strong, if not unrealistic, assumptions.

### A.2 NEW AGGREGATIONS

This section provides additional details on newly proposed gradient aggregation methods: (i) Capped MGDA, which is presented in the main paper; and (ii) Greedy Aggregation with Dual Cone Projection (Greedy-DCP), which we introduce here.

### A.2.1 CAPPED MGDA

Here we derive the dual formulation of Capped MGDA from the CVaR primal formulation:

$$\min_{\mathbf{d},\alpha} \alpha + C \sum_{k=1}^{m} \max \{0, \langle -\mathbf{d}, \mathbf{g}_k \rangle - \alpha\} + \frac{1}{2} \|\mathbf{d}\|^2, \tag{primal}$$

*Proof.* First, as a standard optimization technique, the above is equivalent to

$$\min_{\mathbf{d},\alpha} \max_{0 \le \lambda_k \le C} \alpha + \sum_{k=1}^{m} \lambda_k(\langle -\mathbf{d}, \mathbf{g}_k \rangle - \alpha) + \frac{1}{2} \|\mathbf{d}\|^2 \tag{49}$$

Dual problem is formed by switching the min-max (strong duality holds since the objective is convex in $\mathbf{d}, \alpha$; and linear in $\boldsymbol{\lambda}$ defined on a compact domain), and simplify:

$$\max_{0 \le \lambda_k \le C} \min_{\mathbf{d},\alpha} \alpha + \sum_{k=1}^{m} \lambda_k(\langle -\mathbf{d}, \mathbf{g}_k \rangle - \alpha) + \frac{1}{2} \|\mathbf{d}\|^2 \tag{50}$$

$$\max_{0 \le \lambda_k \le C} \min_{\mathbf{d},\alpha} \alpha \left(1 - \sum_{k=1}^{n} \lambda_k\right) - \left\langle \mathbf{d}, \sum_{k=1}^{n} \lambda_k \mathbf{g}_k \right\rangle + \frac{1}{2} \|\mathbf{d}\|^2 \tag{51}$$

$$\max_{\lambda_k \le C, \boldsymbol{\lambda} \in \Delta} \min_{\mathbf{d}} - \left\langle \mathbf{d}, \sum_{k=1}^{n} \lambda_k \mathbf{g}_k \right\rangle + \frac{1}{2} \|\mathbf{d}\|^2 \tag{52}$$

For the last line, the inner minimization is achieved when $\mathbf{d} = \sum_{k=1}^{n} \lambda_k \mathbf{g}_k$.

Then substitute this in and move the negative sign out, we reach the final dual problem:

$$\min_{\boldsymbol{\lambda} \le C, \boldsymbol{\lambda} \in \Delta} \|\sum_{k=1}^{n} \lambda_k \mathbf{g}_k\|^2, \tag{53}$$

Equivalently, in Jacobian notation:

$$\min_{\boldsymbol{\lambda} \le C, \boldsymbol{\lambda} \in \Delta} \|J\boldsymbol{\lambda}\|^2, \text{ and } \mathbf{d} = J\boldsymbol{\lambda}. \tag{54}$$

$\square$

We call this new aggregation method *Capped MGDA*, since its dual problem closely resembles MGDA, with an additional cap constraint on the coefficients, i.e. $\boldsymbol{\lambda} \leq C$.

While one may view Capped MGDA as a natural extension of the dual formulation of MGDA, we emphasize that its primal CVaR formulation not only offers an intuitive interpretation of the method's objective, but also greatly facilitates the convergence analysis, which would be presumably difficult to establish from the seemingly simpler dual formulation.

### A.2.2 GREEDY AGGREGATION WITH DUAL CONE PROJECTION (GREEDY-DCP)

This method is a *non-conflicting* aggregator that, similar to UPGrad, relies on projecting gradients onto the dual cone as the first step. We first project each gradient $\mathbf{g}_k$ onto the dual $\text{cone}^* J := \{\mathbf{d} : J^\top \mathbf{d} \geq \mathbf{0}\}$, and denote the result by $\mathbf{p}_k$. The greedy aggregation is then formulated as

$$\underset{\mathbf{d}}{\arg\min} \, \min_k \left( -\langle \mathbf{p}_k, \mathbf{d} \rangle + \tfrac{1}{2}\|\mathbf{d}\|^2 \right) \tag{55}$$

By switching the order of minimization, we obtain the algorithmic update:

$$\mathbf{d} = \mathbf{p}_i, \quad \text{where } i = \underset{k}{\arg\max} \|\mathbf{p}_k\|. \tag{56}$$

The convergence of Greedy-DCP follows directly from Corollary 2 established in the main paper.

## B PROOFS OMITTED FROM THE MAIN TEXT

**Theorem 1** (Sufficient Alignment Condition). *Suppose there exists an $L$-smooth function $F : \mathbb{R}^d \to \mathbb{R}_+$ such that the directions $\mathbf{d}_t$ satisfy:*

$$\langle \mathbf{d}_t, \nabla F(\mathbf{w}_t) \rangle \geq c_t \Gamma_t \|\mathbf{d}_t\|, \qquad \text{with } c_t \geq 0. \tag{9}$$

*With suitably chosen step size $\eta_t$ (so that (59) in Appendix B holds; for instance, when $\eta_t = \frac{c_t \Gamma_t}{L\|\mathbf{d}_t\|}$), if $c_t \geq c > 0$, then $\sum_t \Gamma_t^2 \leq \frac{2LF(\mathbf{w}_0)}{c^2}$. In particular, $\min_{t \leq T} \Gamma_t \leq \sqrt{\frac{2LF(\mathbf{w}_0)}{c^2 T}}$ and $\lim_{t \to \infty} \Gamma_t = 0$.*

*Proof.* Applying the descent lemma we have

$$F(\mathbf{w}_{t+1}) \leq F(\mathbf{w}_t) - \eta_t \langle \nabla F(\mathbf{w}_t), \mathbf{d}_t \rangle + \tfrac{L}{2}\|\eta_t \mathbf{d}_t\|^2 \tag{57}$$

$$\leq F(\mathbf{w}_t) - c_t \Gamma_t \|\eta_t \mathbf{d}_t\| + \tfrac{L}{2}\|\eta_t \mathbf{d}_t\|^2 \tag{58}$$

Optimizing $\eta_t = \frac{c_t \Gamma_t}{L\|\mathbf{d}_t\|}$, we obtain

$$F(\mathbf{w}_{t+1}) \leq F(\mathbf{w}_t) - \frac{c_t^2 \Gamma_t^2}{2L}. \tag{59}$$

Telescoping and noting that $F \geq 0$:

$$\sum_t c_t^2 \Gamma_t^2 \leq 2LF(\mathbf{w}_0), \tag{60}$$

whence follows both claims. $\square$

[Optionally] For a weaker conclusion under more relaxed assumption, the same proof also shows that

- if $\sum_t c_t^2 = \infty$, then $\liminf_{t \to \infty} \Gamma_t = 0$,

namely that, there exists a subsequence of $\Gamma_t$ that converges to 0.

**Theorem 2** (Convergence of Non-Conflicting Directions). *If the direction $\mathbf{d}_t \in \text{conv}(J_\mathbf{f}(\mathbf{w}_t))$ and $\mathbf{d}_t \in \text{cone}^*(J_\mathbf{f}(\mathbf{w}_t))$ (i.e., non-conflicting), then condition (A) and hence Corollary 1 holds with $c_t \equiv 1$ and $F = \sum_k f_k$.*

*Proof.* We omit the index $t$ in the following to simplify the notation.

Let $F = \sum_k f_k$ and $\mathbf{d} = J_{\mathbf{f}}(\mathbf{w})\boldsymbol{\lambda}$ for some $\boldsymbol{\lambda} \in \Delta$. We directly verify (A):

$$\langle \mathbf{d}, \nabla F(\mathbf{w}) \rangle = \sum_k \langle \mathbf{d}, \nabla f_k(\mathbf{w}) \rangle \geq \sum_k \lambda_k \langle \mathbf{d}, \nabla f_k(\mathbf{w}) \rangle = \left\langle \mathbf{d}, \sum_k \lambda_k \nabla f_k(\mathbf{w}) \right\rangle = \|\mathbf{d}\|^2, \quad (61)$$

where the inequality is due to $\mathbf{d} \in \text{cone}^*(J_{\mathbf{f}}(\mathbf{w}))$ so that $\langle \mathbf{d}, \nabla f_k(\mathbf{w}) \rangle \geq 0$ and $\lambda_k \in [0,1]$. $\quad\square$

**Proposition 1.** *Let* $\mathbf{q} \in \text{cone}(J)$, *i.e.,* $\mathbf{q} = J\boldsymbol{\mu}$ *for some* $\boldsymbol{\mu} \geq 0$. *Then,* $\mathbf{d} := P_{\text{cone}^*(J)}(\mathbf{q}) \in \text{cone}^*(J) \cap \text{cone}(J)$. *In particular,* $\mathbf{d} = J\boldsymbol{\nu}$ *for some* $\boldsymbol{\nu} \geq 0$ *such that* $\|\boldsymbol{\nu}\|_1 \geq \|\boldsymbol{\mu}\|_1$.

*Proof.* For any (closed) convex cone $K$, we recall Moreau's celebrated decomposition (Moreau, 1962):

$$\mathbf{d} = P_K(\mathbf{q}), \mathbf{d}^* = -P_{K^*}(-\mathbf{q}) \iff \mathbf{d} \perp \mathbf{d}^*, \mathbf{d} + \mathbf{d}^* = \mathbf{q}, \mathbf{d} \in K, \mathbf{d}^* \in -K^*. \quad (62)$$

Thus, with $K = \text{cone}^*(J)$ and hence $K^* = \text{cone}(J)$, we have

$$\mathbf{d} = \mathbf{q} + P_{\text{cone}(J)}(-\mathbf{q}) = J\boldsymbol{\mu} + J\boldsymbol{\alpha}, \quad \text{where} \quad \boldsymbol{\alpha} \geq \mathbf{0}. \quad (63)$$

It follows that we can set $\boldsymbol{\nu} = \boldsymbol{\mu} + \boldsymbol{\alpha}$, and the proof is complete. $\quad\square$

**Theorem 3** ([Convergence under Monotone Descent]). *Suppose each objective $f_k$ is $L$-smooth, convex and bounded from below. Choose $\mathbf{d}_t \in \text{conv}(J_{\mathbf{f}}(\mathbf{w}_t))$ (and step size $\eta_t \equiv \eta \leq \frac{1}{L}$) such that the function values $\{\mathbf{f}(\mathbf{w}_t)\}$ monotonically decrease. Then, there exists $\boldsymbol{\lambda} \in \Delta$ such that the iterates $\mathbf{w}_t$ defined in* (8) *satisfy: for any* $\mathbf{w}$,

$$\langle \boldsymbol{\lambda}, \mathbf{f}(\mathbf{w}_t) \rangle - \langle \boldsymbol{\lambda}, \mathbf{f}(\mathbf{w}) \rangle \leq \frac{1}{2\eta t} \|\mathbf{w}_0 - \mathbf{w}\|^2. \quad (11)$$

*Proof.* From $L$-smoothness, we have

$$f_k(\mathbf{w}_{t+1}) \leq f_k(\mathbf{w}_t) - \eta \langle \mathbf{d}_t, \nabla f_k(\mathbf{w}_t) \rangle + \frac{L\eta^2}{2} \|\mathbf{d}_t\|^2, \quad (64)$$

Since $f$ is convex, for all $\mathbf{w}$:

$$f_k(\mathbf{w}_{t+1}) \leq f_k(\mathbf{w}) + \langle \mathbf{w}_t - \mathbf{w}, \nabla f_k(\mathbf{w}_t) \rangle - \eta \langle \mathbf{d}_t, \nabla f_k(\mathbf{w}_t) \rangle + \frac{L\eta^2}{2} \|\mathbf{d}_t\|^2. \quad (65)$$

Rearranging and simplifying:

$$f_k(\mathbf{w}_{t+1}) - f_k(\mathbf{w}) \leq \langle \mathbf{w}_t - \mathbf{w} - \eta\mathbf{d}_t, \nabla f_k(\mathbf{w}_t) \rangle + \frac{L\eta^2}{2} \|\mathbf{d}_t\|^2. \quad (66)$$

Taking inner product with $\boldsymbol{\lambda}_t$ on both sides:

$$\langle \boldsymbol{\lambda}_t, \mathbf{f}(\mathbf{w}_{t+1}) - \mathbf{f}(\mathbf{w}) \rangle \leq \langle \mathbf{w}_t - \mathbf{w} - \eta\mathbf{d}_t, \mathbf{d}_t \rangle + \frac{L\eta^2}{2} \|\mathbf{d}_t\|^2 = \langle \mathbf{w}_t - \mathbf{w}, \mathbf{d}_t \rangle + (\frac{L\eta^2}{2} - \eta)\|\mathbf{d}_t\|^2. \quad (67)$$

As long as $\eta \leq \frac{1}{L}$, the above implies

$$\langle \boldsymbol{\lambda}_t, \mathbf{f}(\mathbf{w}_{t+1}) - \mathbf{f}(\mathbf{w}) \rangle \leq \frac{1}{2\eta}(\|\mathbf{w}_t - \mathbf{w}\|^2 - \|\mathbf{w}_t - \mathbf{w} - \eta\mathbf{d}_t\|^2) \quad (68)$$

$$= \frac{1}{2\eta}(\|\mathbf{w}_t - \mathbf{w}\|^2 - \|\mathbf{w}_{t+1} - \mathbf{w}\|^2). \quad (69)$$

Telescoping we arrive at:

$$\frac{1}{T} \sum_{t=0}^{T-1} \langle \boldsymbol{\lambda}_t, \mathbf{f}(\mathbf{w}_{t+1}) - \mathbf{f}(\mathbf{w}) \rangle \leq \frac{\|\mathbf{w}_0 - \mathbf{w}\|^2}{2\eta T} \quad (70)$$

Since $\mathbf{f}(\mathbf{w}_t)$ monotonically decreases, we further lower bound the left-hand side:

$$\frac{1}{T} \sum_{t=0}^{T-1} \langle \boldsymbol{\lambda}_t, \mathbf{f}(\mathbf{w}_T) - \mathbf{f}(\mathbf{w}) \rangle \leq \frac{\|\mathbf{w}_0 - \mathbf{w}\|^2}{2\eta T} \quad (71)$$

Denoting $\boldsymbol{\lambda} := \frac{1}{T} \sum_{t=0}^{T-1} \boldsymbol{\lambda}_t$, we conclude that for all $\mathbf{w}$:

$$\langle \boldsymbol{\lambda}, \mathbf{f}(\mathbf{w}_T) \rangle - \langle \boldsymbol{\lambda}, \mathbf{f}(\mathbf{w}) \rangle \leq \frac{\|\mathbf{w}_0 - \mathbf{w}\|^2}{2\eta T}. \tag{72}$$

The proof is now complete. $\qquad\square$

To exploit (72) as tightly as possible, we simply take $\mathbf{w} = \mathbf{w}_* = \operatorname{argmin}_{\mathbf{w}} \langle \boldsymbol{\lambda}, \mathbf{f}(\mathbf{w}) \rangle$, which is weakly Pareto optimal for convex $\mathbf{f}$ (and Pareto optimal for strictly convex $\mathbf{f}$). Thus, the iterate $\mathbf{w}_t$ converges at rate $O(1/t)$ in terms of the $\boldsymbol{\lambda}$-averaged function value.

Furthermore, note that $\boldsymbol{\lambda}$ is in the simplex, a compact set. Thus, it possesses a convergent subsequence with limit $\boldsymbol{\lambda}_\star$. Assuming that the corresponding iterates are bounded, then passing to the limit in inequality (72) shows that every accumulation point $\mathbf{w}_\star$ is weakly Pareto optimal, since $\mathbf{w}_\star$ minimizes the scalarized objective $\langle \boldsymbol{\lambda}_\star, \mathbf{f} \rangle$. Additionally, when $\boldsymbol{\lambda}_\star \in \operatorname{ri}(\Delta)$, $\mathbf{w}_\star$ is actually Pareto optimal.

**Remark.** For non-conflicting directions, monotonicity is guaranteed provided the descent direction $\mathbf{d}$ lies in the interior of the dual cone $\operatorname{cone}^*(J_\mathbf{f}(\mathbf{w}))$ (as in MGDA, UPGrad, or Nash-MTL) and the step size $\eta$ is chosen appropriately. In degenerate cases where $\mathbf{d}$ lies on the boundary of $\operatorname{cone}^*(J_\mathbf{f}(\mathbf{w}))$, additional mechanisms such as line search or perturbation can be adopted to maintain monotonicity.

To see this, apply the L-smooth inequality for all $f_k$, we have:

$$f_k\left(\mathbf{w}_t - \eta_t \mathbf{d}_t\right) \leq f_k\left(\mathbf{w}_t\right) - \eta_t \left\langle \nabla f_k\left(\mathbf{w}_t\right), \mathbf{d}_t \right\rangle + \frac{L}{2}\eta_t^2 \left\|\mathbf{d}_t\right\|^2, \; \forall k \tag{73}$$

Using the strengthened non-conflicting condition,

$$\langle \nabla f_k(\mathbf{w}_t), \mathbf{d}_t \rangle > 0, \; \forall k \tag{74}$$

and choosing

$$0 < \eta_t \leq \frac{\min_k \langle \nabla f_k(\mathbf{w}_t), \mathbf{d}_t \rangle}{L \left\|\mathbf{d}_t\right\|^2} \tag{75}$$

yields $\mathbf{f}(\mathbf{w}_t - \eta_t \mathbf{d}_t) \leq \mathbf{f}(\mathbf{w}_t)$.

## C  EXPERIMENT DETAILS

### C.1  NON-CONFLICTING GRADIENT AGGREGATORS

#### C.1.1  METHODS

**Existing aggregators.** Whenever possible, we stick to the official implementations of all methods, and otherwise use the TorchJD repository (Quinton & Rey, 2024) as a reference. For Nash-MTL (Navon et al., 2022), we adopt the official implementation's default, which always clips the aggregated update direction to satisfy $\|\mathbf{d}_t\| = 1$. All examined methods are run in their deterministic, full-batch form, without momentum. For the normalized variants (e.g., Nash-MTL*, UPGrad*, DualProj*), we keep the original implementations and simply re-scale $\mathbf{d}_t$ to lie in the convex hull of gradients $\{\mathbf{g}_k\}$ by normalizing the weighting coefficients, $\boldsymbol{\lambda}_t \leftarrow \frac{\boldsymbol{\lambda}_t}{\|\boldsymbol{\lambda}_t\|_1}$, ensuring that $\boldsymbol{\lambda}_t \in \Delta$.

**Mixed Aggregator Scheduling (MAS).** For MAS, we consider two schedulings: (1) Uniform random selection ('Rand'), where each iteration one aggregator is randomly chosen from the pool {MGDA, Nash-MTL*, UPGrad*, DualProj*}; (2) Round-robin every $n$ iterations ('RR($n$)'), where each aggregator in the pool is applied for $n$ consecutive iterations before switching to the next.

**Misc.** The learning rate $\eta$ used for synthetic problems is 0.001, while for fairness classification benchmark is 0.005.

### C.1.2 SYNTHETIC PROBLEM DETAILS

Here we provide explicit definitions for VLMOP2 and Omnitest objectives used in our paper.

**VLMOP2 (van Veldhuizen & Lamont, 1999).**

$$\min_{\mathbf{x} \in \mathbb{R}^n} \quad f_1(\mathbf{x}) = 1 - \exp\left(-\sum_{i=1}^n \left(x_i - \tfrac{1}{\sqrt{n}}\right)^2\right), \tag{76}$$

$$f_2(\mathbf{x}) = 1 - \exp\left(-\sum_{i=1}^n \left(x_i + \tfrac{1}{\sqrt{n}}\right)^2\right), \tag{77}$$

$$\text{s.t.} \quad -2 \le x_i \le 2, \quad i = 1, \ldots, n. \tag{78}$$

**Omnitest (Deb & Tiwari, 2008).**

$$\min_{\mathbf{x} \in \mathbb{R}^n} \quad f_1(\mathbf{x}) = \sum_{i=1}^n \sin(\pi x_i), \tag{79}$$

$$f_2(\mathbf{x}) = \sum_{i=1}^n \cos(\pi x_i), \tag{80}$$

$$\text{s.t.} \quad 0 \le x_i \le 6, \quad i = 1, \ldots, n. \tag{81}$$

For both problems, although $\mathbf{x}$ is formally constrained, we initialize it in the interior and ensure that the entire trajectory—including the Pareto solution it converges to—remains in the interior. This allows us to empirically treat them as unconstrained MOO problems and apply all the gradient aggregation methods considered in this paper.

### C.1.3 FAIRNESS CLASSIFICATION

**Setup.** For the fairness classification task on the **Adult** dataset, we follow the LibMOON benchmark (Zhang et al., 2024) for both dataset preprocessing and model architecture. Specifically, we use the function `libmoon.util.mtl.get_dataset("adult")` to generate the train, validation, and test splits. For the model, we adopt LibMOON's `M4 fair_model` architecture: a fully connected neural network consisting of three hidden layers of dimension 256 each, with ReLU activations. The binary classification task is whether the annual income is greater than \$50K.

We use the Difference of Equalized Odds (DEO) as our fairness metrics. Following Hardt et al. (2016), we define

$$\text{DEO1} = \left| \Pr\{\widehat{Y} = 1 \mid A = 0, Y = 1\} - \Pr\{\widehat{Y} = 1 \mid A = 1, Y = 1\} \right|, \tag{82}$$

$$\text{DEO2} = \left| \Pr\{\widehat{Y} = 1 \mid A = 0, Y = 0\} - \Pr\{\widehat{Y} = 1 \mid A = 1, Y = 0\} \right|. \tag{83}$$

where $\widehat{Y}$ denotes the model's predicted label, $A$ the sensitive attribute (e.g., gender), and $Y$ the ground-truth label. To make these metrics differentiable, we apply a `tanh` relaxation that replaces the indicator function $\mathbf{1}\{p_i \ge 0.5\}$, yielding smooth surrogate losses. Thus, the three objectives are: $f_1$, the binary cross-entropy loss; $f_2$, the relaxed DEO1; and $f_3$, the relaxed DEO2.

**More results.** Here we provide additional experimental results for fairness classification; see Figure 5 and Figure 6.

We observe that the Pareto stationarity measure $\gamma(\mathbf{w}_t)$ converges to $0$ for all methods except Nash-MTL (without normalization), which suffers from overshooting because $\|\mathbf{d}\|$ is fixed at $1$, leading to instability near Pareto stationarity. Applying convex-hull normalization to Nash-MTL yields smoother convergence, and a similar but less pronounced effect is also observed for UPGrad.

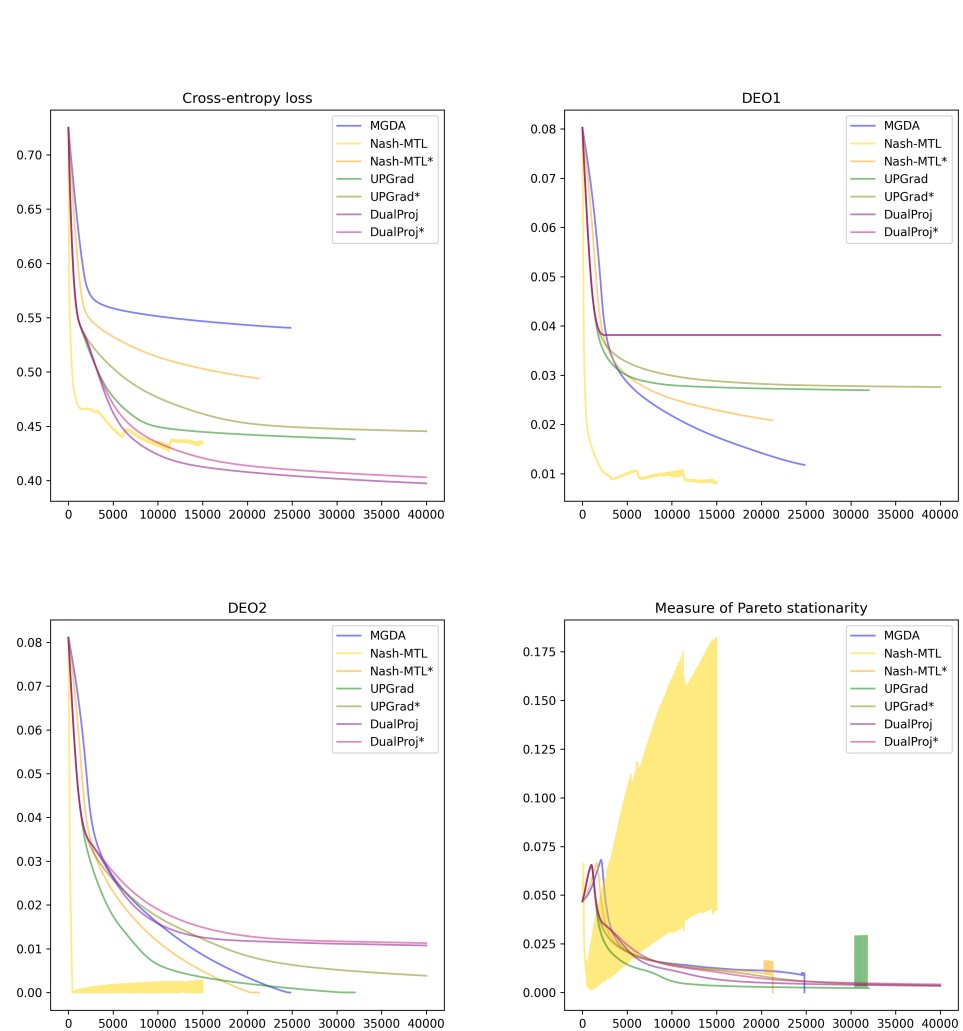

Figure 5: Non-conflicting aggregators on LibMOON fairness classification benchmark. Nash-MTL is unstable, while Nash-MTL* (the normalized variant) is smooth and stable.

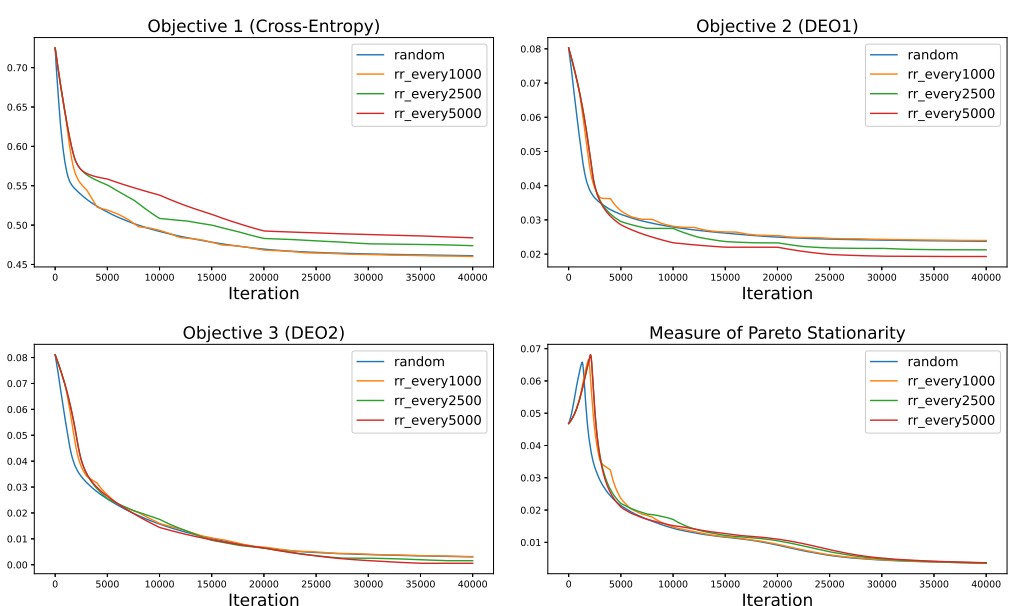

Figure 6: Mixed aggregator scheduling on fairness classification benchmark.

## C.2 CAPPED MGDA EXPERIMENTS DETAIL

**Data.** We conduct experiments on the CIFAR-10 dataset in the adversarial federated learning setting. To create a non-i.i.d. partition, we follow the sampling procedure of Hu et al. (2022). Specifically, we first sort all data points by class and then split them consecutively into 250 shards of 200 images each, where each shard contains images from a single class. Each client is randomly assigned 10 distinct shards, resulting in 2000 instances per client. This produces heterogeneous data distributions across clients, as each client has access to different subsets of class labels. For example, Client 0's data includes class labels $\{0, 3, 4, 5, 6, 7, 9\}$ while lacking labels $\{1, 2, 8\}$.

**Model.** We use the same lightweight CNN for MGDA, Capped-MGDA, and the centralized-training baseline; see Table 3 for the configuration.

Table 3: Network architecture for CIFAR-10 experiments with CNN.

| Layer Type | Configuration | Activation |
|---|---|---|
| Conv2D | $3 \times 32$, kernel size 5 | ReLU |
| BatchNorm2D | 32 channels | – |
| MaxPool2D | kernel size 2, stride 2 | – |
| Conv2D | $32 \times 32$, kernel size 5 | ReLU |
| BatchNorm2D | 32 channels | – |
| MaxPool2D | kernel size 2, stride 2 | – |
| Fully Connected | $32 \times 5 \times 5 \rightarrow 384$ | ReLU |
| Fully Connected | $384 \rightarrow 192$ | ReLU |
| Fully Connected | $192 \rightarrow 10$ | – |

**Adversarial FL setting.** In the adversarial federated learning setup, during each epoch's gradient aggregation a single malicious attacker injects a crafted gradient that opposes one randomly selected participant's gradient. Concretely, letting $k$ be sampled uniformly from the clients, the injected gradient is $\mathbf{g}_{\mathrm{adv}} = -\mathbf{g}_k + \epsilon \mathcal{N}(0, I)$, with noise scale $\epsilon = 0.01$.

**Misc.** We use a learning rate of $\eta = 0.01$, train for 1000 epochs, and adopt full-batch training to ensure determinism.

### C.2.1 ADDITIONAL RESULTS ON CAPPED MGDA AND THE ADVERSARIAL FEDERATED LEARNING SETTING

In this subsection, we provide additional empirical results to further illustrate the behavior of Capped MGDA and other methods under adversarial federated learning. These results complement the main experiments in Section 5.2 by offering finer-grained insights as well as comparisons with related variants such as MGDA with clipping.

Table 4 reports the final global test accuracy across a broader set of aggregation methods. Empirically, we observe that MGDA performs poorly under adversarial gradients, whereas projection-based methods such as UPGrad* and DualProj* remain substantially more robust, with NashMTL* showing moderate performance. Notably, Capped MGDA with a small cap (i.e. $C = 0.1$) achieves accuracy comparable to the strongest baselines, indicating that restricting the influence of the adversarial client substantially improves robustness.

Figure 7 compares Capped MGDA with a naive "MGDA + coefficient-clipping" variant using the same threshold. The performance gap highlights that simply clipping the MGDA coefficients is insufficient; the constrained optimization formulation underlying Capped MGDA is non-trivial and essential for producing effective descent directions.

Figure 8 visualizes average client accuracy during training, for a range of cap values $C$. We observe a consistent trend: smaller $C$ leads to higher robustness and improved accuracy. This aligns with the intuition that a tighter cap limits the adversarial client's ability to distort the aggregated gradient.

As $C$ decreases, the aggregated direction becomes more stable across clients, leading to smoother training dynamics and better overall performance.

Table 4: Comparison of global test accuracy across methods under the adversarial federated learning setting. All values report mean accuracy of 10 clients after 1000 training epochs.

| Method | Global Test Accuracy |
| --- | --- |
| No adversary baseline | 0.541 |
| MGDA | 0.211 |
| UPGrad* | 0.511 |
| DualProj* | **0.515** |
| NashMTL* | 0.433 |
| Capped MGDA ($C = 0.1$) | **0.515** |

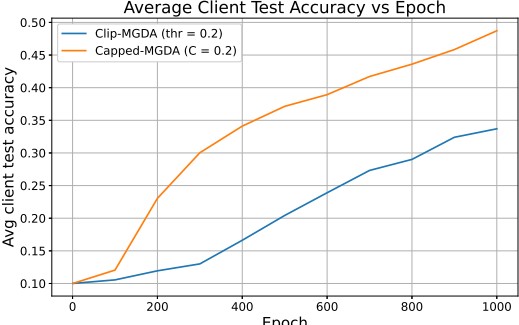

Figure 7: Capped MGDA vs 'MGDA + coefficient clipping'. We observe the later is not as effective, given the same threshold 0.2.

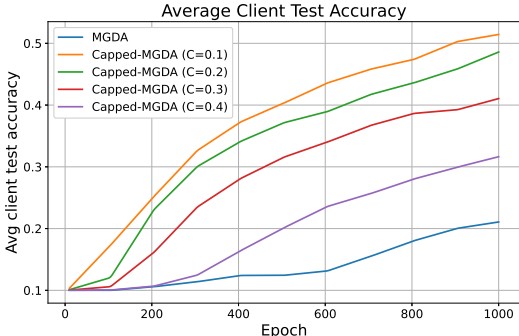

Figure 8: Average client accuracy vs Epoch. We can see a clear improvement when $C$ becomes smaller, which means more robustness by limiting the impact of adversarial gradient.