# OpenReview forum: "A Unifying Framework for Gradient Aggregation in Multi-Objective Optimization"
_ICLR.cc/2026/Conference — Submitted to ICLR 2026_

### Official Review · Reviewer_gatF · 2025-10-28

**Soundness:** 3
**Presentation:** 3
**Contribution:** 3
**Rating:** 6
**Confidence:** 5

**Summary:**

This paper proposed a unified gradient-aggregation multi-objective framework, covering multiple existing MOO methods, and providing a simplified convergence analysis. The core criterion of this framework is the sufficient alignment condition. This condition states that as long as the update direction aligns well with the gradient of a fixed surrogate function, such as the sum of objective functions, then it is guaranteed to converge to a Pareto stationary point.

Additionally, Theorem 2 and Theorem 4 provide two ways to get the desired update $d_t$. The former requires a non-conflicting direction, and the latter allows conflict through the construction of the primal subproblem. This framework not only serves as an analysis tool for existing methods, but also provides guidelines for designing new MOO methods. Following this guideline, this paper proposes capped MGDA, which is more robust than vanilla MGDA.

Experiments on the existing methods validate that Pareto stationary points can be achieved, which is consistent with the framework analysis. Experiments on the federated learning setting demonstrate the robustness of capped MGDA, further demonstrating that the proposed framework can provide practical guidelines to design new MOO methods.

**Strengths:**

1. This paper proposes a clear and reusable framework, providing not only simplified convergence analysis for existing MOO methods, but also guidelines for designing new variants.
2. This paper provides a comprehensive convergence analysis, including convex and non-convex cases.
3. Experiments on both synthetic and realistic benchmarks demonstrate the feasibility of the framework.

**Weaknesses:**

1. The selection of $\Gamma_t$ should be clarified. Line 201,206,209 suggest different choices. Are there any requirements or constraints for $\Gamma_t$? Theorem 1 provides little information on this aspect, which makes the explanation confusing.
2. The discussion of power-mean-based directions in Section 4.3 is limited. The harmonic mean case (when $p=-1$) is closely related to the minimum potential delay fairness used in FairGrad [1] and PIVRG [2].

3. Experiments in Section 5.1 are somewhat trivial. The goal is to show existing methods converge to Pareto stationary points, thus demonstrating consistency with the theoretical results of the proposed framework. However, the convergence of most existing methods has already been discussed in their original papers. Therefore, it would be better to provide more comparisons across these methods.
Table 1 shows that these methods solve different subproblems by using different $s(x),r,q_t$. Figure 2 and 3 present different learning behaviors of these methods, even though they all achieve Pareto stationarity. So, it would be helpful to explain these different behaviors from the perspective of their corresponding subproblems.

4. Experiments in Section 5.2 validate that the proposed framework can be used as practical guidelines to design new MOO variants. The capped MGDA replaces MGDA’s maximizing the worst task progress $\langle g_k, d \rangle$, with a CVaR-style objective that emphasizes tasks whose progress is below a threshold $\alpha$. So, it is somewhat expected to perform more robustly than MGDA under adversarial attack. To isolate why it helps, it would be better to compare MGDA + cap constraints, as well as to provide more details on hyperparameter selection.


[1] Fair Resource Allocation in Multi-Task Learning. [ICML 2024]

[2] Revisiting Fairness in Multitask Learning: A Performance-Driven Approach for Variance Reduction. [CVPR 2025]

**Questions:**

See the weaknesses.

---

> ### Author Response · Authors · 2025-11-21
> **Thank you for your insightful comments and we answer your questions below.**
>
> - **1. Clarify the choice and requirements of $\Gamma_t$.**
>
>   We thank the reviewer for pointing out the ambiguity regarding the choice of $\Gamma_t$. We apologize for the confusion—our intention in presenting Theorem 1 in a general form was to keep it flexible so that it could be reused in different settings (including future work by others) without committing to a single choice of $\Gamma_t$. In theory, $\Gamma_t$ is a user-defined 'quantity of interest' reflecting some measure related to convergence, and the only requirement for it is __positivity__. The benefit of leaving $\Gamma_t$ unspecified in (9) is that Theorem 1 can then be instantiated for different purposes:
>   - Single-objective case: Theorem 1 can reduce to the angle constraint, which we use in Appendix A.1.8 to recover the convergence guarantee of CAGrad in the regime $0 \leq c <1$.
>   - Multi-objective case (naively): A natural choice under the multi-objective setting is $\Gamma_t = \gamma(\mathbf{w}_t)$, the measure of Pareto stationarity. However, directly substituting this into (9) yields a condition that can be difficult to check and/or apply in practice.
>   - Multi-objective case (main application in this paper): To obtain a condition that is both practical and broadly applicable, we instantiate Theorem 1 with: $$\Gamma_t = \|\mathbf{d}_t\|,$$ which upper-bounds $\gamma$ (when the direction lies in the convex hull) and leads directly to the __alignment condition (A)__ used throughout our analysis. This choice makes (9) easy to verify and thus is used to establish later theoretical guarantees in the paper (e.g. Theorem 2, Theorem 4).
>
>   To avoid confusion, we revise the paper to explicitly suggest that for understanding our main results, one may simply view $\Gamma_t$ as $\|\mathbf{d}_t\|$, while Theorem 1 is kept in its general form to allow broader applicability.
>
> - **2. Expand discussion of the harmonic mean case, including FairGrad and PIVRG.**
>
>   We thank the reviewer for bringing these relevant works to our attention: we have cited them in the revised Section 4.3.1, added discussion regarding convergence guarantees of the two methods, and how to apply our framework to them in Appendix A.1.4 and A.1.5, respectively.
>
> - **3. Provide more insights regarding Section 5.1 results (Fig 2 and 3) and discuss convergence behaviors of these methods.**
>
>   Thank you for the suggestion. We have expanded the discussion of the empirical results in Section 5.1 (highlighted in blue). The key observations are:
>   - (1) For Figure 2, all the non-conflicting methods exhibit similar asymptotic behavior when comparing their normalized ("*") variants; the unnormalized versions only appear faster because their larger $\|\mathbf{d}_t\|$ effectively results in larger step sizes (e.g. Nash-MTL in particular).
>   - (2) Figure 3 confirms the validity of MAS: switching among different non-conflicting methods within a single optimization run, surprisingly, still ensures convergence to Pareto-stationarity (with similar end-phase asymptotics), consistent with the minimal requirement implied by Theorem 2.
>
>   Additionally, from a theoretical perspective, we have substantially expanded the discussion in Appendix A.1 (highlighted in blue) to cover the convergence guarantees of most existing methods, both as originally analyzed in their respective papers and as reinterpreted through our framework. While it is true that many of these methods have convergence analyses, our unified framework can yield stronger guarantees under weaker assumptions (e.g., establishing a convergence rate for Nash-MTL) and greatly simplify the analyses for a wide range of methods. We hope this expanded discussion provides a more unified perspective on the convergence properties of existing methods.

---

> ### Author Response · Authors · 2025-11-21
>
> - **4. Section 5.2. Compare MGDA with cap constraints and provide more details on hyperparameter selection for capped MGDA.**
>
>   Thank you for the suggestion. In the revised version, we include additional results for capped MGDA in the adversarial FL setting using a range of capping values $C$ to illustrate its interpolation behavior. We also add a figure that visualizes the “non-conflictingness" of $\mathbf{d}_t$ throughout optimization [see Figure 4 (Bottom Right)]. Our observations reveal a clear trade-off between non-conflictingness and update norm: bigger values of $C$ make the method behave more like MGDA, with stronger non-conflictingness but producing smaller update norms, which in turn leads to weaker final performance. Conversely, smaller values of $C$ cause the method to behave more like linear scalarization, resulting in larger update norms but reduced non-conflictingness.
>
>   Regarding the suggestion to “compare MGDA + cap constraints,” we interpret this as a request for clarification on how capped MGDA differs from a post-hoc clipping strategy. First, we emphasize that capped MGDA is derived from a _primal_ CVaR-style formulation and leads to the _dual_ problem $$\min_{\mathbf{\lambda} \leq C, \mathbf{\lambda} \in \Delta} \|\sum_{k=1}^n \lambda_k \mathbf{g}_k\|^2$$ as shown in Sec. 4.3 (Eq. 23) and Appendix A.2 (Eq. 53). Thus, capped MGDA solves an MGDA-like quadratic program, but with explicit cap constraints (i.e. $\mathbf{\lambda} \leq C$) built into the optimization problem itself. If the reviewer instead meant applying MGDA first and then clipping the resulting coefficients $\lambda_k$ to satisfy $\lambda_k \leq C$, we have tested this heuristic and found it significantly less effective (see Appendix C.2.1). The limitation can be understood through a simple example: if we have three gradients where two are nearly opposite (e.g., one adversarial), MGDA assigns each of the opposite gradients a weight of $\frac{1}{2}$. Clipping these weights still yields a nearly cancelling combination, resulting in $\mathbf{d}_t \approx 0$. In contrast, capped MGDA avoids this degeneracy precisely because the cap constraint is integrated into the optimization, not applied afterward.

---

> ### Author Response · Authors · 2025-11-27
> **Follow-Up on Rebuttal Discussion**
>
> Dear Reviewer gatF,
>
> As the discussion period is nearing its end, we would like to kindly follow up to see if you have any further feedback after our first round revision and rebuttal. We greatly appreciate your expert evaluation and your positive assessment, and we would be happy to clarify or expand on any remaining aspect of the paper.
>
> Thank you again for your careful and insightful review.
>
> Best regards,
> The Authors

---

### Official Review · Reviewer_MKzs · 2025-10-31

**Soundness:** 3
**Presentation:** 2
**Contribution:** 2
**Rating:** 4
**Confidence:** 4

**Summary:**

This paper proposes a unified framework for MOO algorithms that non-conflicting directions, when chosen within the convex hull of gradients, form a fundamental sufficient condition for convergence. This framework guides further MOO algorithm design.

**Strengths:**

1. The unified framework for MOO is critical and novel.
2. The experiments validate the theorem somehow.

**Weaknesses:**

**Minor issues**
1. Some notations are not clearly defined. For example, $\lambda$ is first defined as a simplex vector. However, in the eq. (7), the $\lambda$ is reused in the definition of the cone, and it is not a simplex. Another new notation can be used here for better clarification.

**Theoretical analysis issues**

2. In Theorem 1, to prove $\lim_{t\rightarrow\infty} \mathcal{T_t}=0$, a condition $\sum_{t}c_t^2=\infty$ should be added. I saw this in the appendix in line 926, but this statement is missing in Theorem 1.
3. In Theorem 2, the formulation of $F$ should be mentioned. Again, $ F=\sum_k f_k$ is shown in the appendix, but missing in Theorem 2. Moreover, Theorem 2 is not rigorous, and the result $C_t\equiv1$ cannot be derived. From the proof, it is true that $\langle d_t,\nabla F (w_t)\rangle\geq\\|d_t\\|^2$. To derive $C_t\equiv1$, another part $\langle d_t,\nabla F (w_t)\rangle\leq\\|d_t\\|^2$ is needed. Or $C_t$ can be any value in the range $(0,1)$.
4. In Theorem 3, the statement says "a proper $d_t$ allows function values ${f(w_t)}$ monotonically decrease". I have to admit that I cannot see the reason, and there is no proof for it. For a single task case, it can be guaranteed, but does it hold for multi-task cases? If not, the derivation will be incorrect.
5. Can theorems be directly applied in stochastic settings? What will be the convergence rate?

**Framework issues**

6. What is the connection between $F$ and ${f}$? I am aware that $F$ can be a sum/weighted sum of ${f}$, but the format will depend on the algorithm. For those dynamic weighting methods, such as MGDA, Nash-MTL, etc, the weights are changing, and can these methods be unified in the framework?

**Experiment issues**

7. In Figure 2, the convergence rates for different methods vary. Can authors have an explanation for it?

**Questions:**

Please check the weakness part.

---

> ### Author Response · Authors · 2025-11-21
> **Thank you for your detailed feedback. We have addressed all your questions below, especially regarding theoretical analysis and comprehensiveness of our framework.**
>
> - **0. Primary clarification: does the framework unify dynamic weighting methods (MGDA, Nash-MTL, etc.)?**
>
>   This is an important question, and we want to state clearly that __yes__—our framework unifies the dynamic weighting methods mentioned (e.g., MGDA, Nash-MTL, etc.). In fact, establishing this unified view is the central goal of this paper. As summarized on the right-hand side of Figure 1, each method is covered by either Theorem 2 (non-conflicting aggregation), Corollary 2 (dual-cone projection) or Theorem 4 (subproblem-based aggregation).
>
>   Briefly, the reason these dynamic weighting methods fit into our framework is that for each of them, one can choose an appropriate surrogate $F$ together with $c_t$ so that condition (A) holds at every iteration, and $F$ __need not__ relate to the (dynamic) coefficients at that iteration. This is precisely what enables the unified convergence analysis across these methods. *Note that we are not claiming these dynamic weighting methods are minimizing a static function $F$. Instead, $F$ is merely a theoretical construct that we exploit to easily prove the convergence of these methods.*

---

> ### Author Response · Authors · 2025-11-21
>
> - **1. Clarify notation of $\mathbf{\lambda}$.**
>
>   We thank the reviewer for the suggestion. In the revised manuscript (highlighted in blue), we now use $\mu$ to denote the coefficients associated with the convex cone, and reserve $\mathbf{\lambda}$ for the coefficients associated with the convex hull.
>
> - **2. Missing condition in Theorem 1?**
>
>   No, Theorem 1 presented in the main paper already assumes $c_t \geq c > 0$ (note $c$ is some positive constant), which guarantees $\sum_t c_t^2 = \infty$.
>
>   The statement previously on Line 926 (now Line 1126) of the appendix is an __optional__ additional result derived under a more relaxed assumption and therefore yields a weaker conclusion (subsequence convergence). It is __not__ used for any subsequent results in the main paper. We have added clarification in the appendix to aviod confusion.
>
> - **3. Regarding Theorem 2, mention the formulation of $F$ and is Theorem 2 rigorous?**
>
>   Thanks for the question. We have updated the statement of Theorem 2 and included the specification of $F$ in the main paper for improved clarity.
>
>   We do __not__ see any issue in Theorem 2 regarding $c_t$: in Theorem 2, we proved that
>   $$
>   \langle \mathbf{d}_t, \nabla F(\mathbf{w}_t) \rangle \geq  \|\mathbf{d}_t\|^2.
>   $$
>   which means condition (A) holds with $c_t=1$, i.e.,
>   $$
>   \langle \mathbf{d}_t, \nabla F(\mathbf{w}_t) \rangle \geq 1 \cdot \|\mathbf{d}_t\|^2.
>   $$
>   Note that we never need the other direction of the inequality. The confusion may be that if the above inequality holds for $c_t = 1$, then it also holds for (say) $c_t = \tfrac12$. This is true and is not contradictory with our result: Theorem 2 says we can pick $c_t = 1$ to satisfy condition (A), but it does not and need not exclude other choices of $c_t$.
>
> - **4. Monotonic decrease of $f(\mathbf{w}_t)$ in Theorem 3.**
>
>   We thank the reviewer for raising this point. To clarify, the 'monotonic decrease' is a condition on $\mathbf{d}_t$ that is required for the theorem to apply, which is _not_ claimed to hold universally for all multi-objective methods. It is essentially a slight strengthening of the non-conflicting property:
>   $$
>   \langle \mathbf{d}_t, \nabla f_k(\mathbf{w}_t) \rangle \geq 0, ~\forall ~k = 1, \ldots, m,
>   $$
>   where the inequalities need to be always strict.
>
>   As noted in Lines 261–263 and in the last paragraph of Appendix B, Theorem 3 is therefore a conditional result: it applies to methods whose update directions satisfy this monotone-descent requirement (including MGDA, Nash-MTL, UPGrad, and several others), but naturally does not apply to methods whose directions violate it.
>
>   To see why strengthening the inequality in non-conflictingness suffices, apply the L-smooth inequality for all $f_k$, we have: $$f_k\left(\mathbf{w}_t-\eta_t \mathbf{d}_t\right) \leq f_k\left(\mathbf{w}_t\right)-\eta_t\left\langle\nabla f_k\left(\mathbf{w}_t\right), \mathbf{d}_t\right\rangle+\frac{L}{2} \eta_t^2\left\|\mathbf{d}_t\right\|^2, ~\forall k$$ By the strengthened non-conflicting condition: $\left\langle\nabla f_k(\mathbf{w}_t), \mathbf{d}_t\right\rangle >0, ~\forall k$,
>   and choosing
>   $$0 < \eta_t \leq \frac{\min_k \left\langle\nabla f_k(\mathbf{w}_t), \mathbf{d}_t\right\rangle}{L\left\|\mathbf{d}_t\right\|^2},$$ yields $\mathbf{f}(\mathbf{w}_t - \eta_t \mathbf{d}_t) \leq \mathbf{f}(\mathbf{w}_t)$.

---

> ### Author Response · Authors · 2025-11-21
>
> - **5. About theoretical results in the stochastic setting.**
>
>   We appreciate the reviewer’s question. Extending our theoretical results to the stochastic setting is indeed important but also substantially more challenging, with well-known issues such as biased descent directions and the need for additional assumptions to establish convergence ([1][2][3]). Unlike the deterministic case, stochastic updates introduce gradient noise that affects both the non-conflicting condition and the alignment guarantees required in our analysis. Handling these effects would require new stability assumptions and considerably more elaborate arguments, which are beyond the scope of the present work.
>
>   Our current work provides, to the best of our knowledge, the first deterministic framework that unifies and strengthens the convergence analyses of many existing multi-objective gradient aggregation methods. We view this as a necessary first step before tackling the stochastic case. As noted in the conclusion, extending the framework to the stochastic setting is a promising and feasible direction, and we plan to pursue it in future work.
>
> - **6. Connection between $F$ and $f$, and whether dynamic weighting methods are unified in the framework?**
>
>   We clarify that $F$ is introduced as a surrogate function to facilitate the analysis; it does not imply that the method must explicitly optimize $F(\mathbf{w}_t)$.  The beauty of our framework is that even for dynamic weighting methods (such as MGDA, Nash-MTL, UPGrad), we can still find an appropriate $F$ together with $c_t$ such that condition (A) holds at every iteration, which leads to convergence guarantee. To elaborate:
>   - __Theorem 2__ applies to all __non-conflicting__ aggregation methods, which includes MGDA, Nash-MTL and UPGrad. In our general proof we take $F= \sum_k f_k$, though other method-specific $F$ can also work, e.g. MGDA with $F=f_k, \forall k$.
>   - __Theorem 4__ applies to all methods with subproblem of the form Eq. (12) where $s$ is decreasing and convex, and $r=\frac{1}{2}\|\cdot\|^2$. In this case, the appropriate surrogate is $F = \langle-\nabla s(\mathbf{0}), \mathbf{f}\rangle$.
>
>   Finally, using a weighted sum of $f_k$ as $F$ is natural: it preserves the L-smoothness and lower-boundedness of the original objectives, making it suitable as a surrogate function for the analysis. However, other choices of $F$ remain possible and may be explored in future work.
>
>
> [1] Liu, S., & Vicente, L. N. (2021). "The stochastic multi-gradient algorithm for multi-objective optimization and its application to supervised machine learning". Annals of Operations Research, vol. 339, pp. 1119–1148.
>
> [2] Fernando, H. et al. (2023). "Mitigating Gradient Bias in Multi-objective Learning: A Provably Convergent Approach". In International Conference on Learning Representations.
>
> [3] Chen, L., Fernando, H., Ying, Y., & Chen, T. (2023). Three-way trade-off in multi-objective learning: Optimization, generalization and conflict-avoidance. _Advances in Neural Information Processing Systems_, 36, 70045-70093.

---

> ### Author Response · Authors · 2025-11-21
>
> - **7. The rates for different methods in Figure 2 vary?**
>
>   Yes and no. In Figure 2, we evaluate four non-conflicting gradient aggregation methods using their original directions $d_t$ and their __normalized__ counterparts (denoted by "*", whose $d_t$ is rescaled to be in the convex hull). All methods use the same update rule $w_{t+1}=w_t-\eta d_t$ (but different $d_t$) with a fixed $\eta$. The normalized variants exhibit similar asymptotic behavior, as expected. The unnormalized variants appear to converge faster because their $\|d_t\|$ can be large—effectively using larger step sizes. This effect is particularly pronounced for Nash-MTL, whose $\|d_t\|$ remains constant even when near stationarity.

---

> ### Author Response · Authors · 2025-11-27
> **Follow-Up on Rebuttal Discussion**
>
> Dear Reviewer MKzs,
>
> As the discussion period is nearing its end, we wanted to kindly follow up to see if you have any additional feedback, particularly on our clarifications to the theoretical analysis and the comprehensiveness of the framework. We appreciate your detailed reading of our proofs and are happy to elaborate further if anything remains unclear.
>
> Thank you again for your thoughtful and thorough review.
>
> Best regards,
> The Authors

---

### Official Review · Reviewer_agTX · 2025-11-01

**Soundness:** 2
**Presentation:** 1
**Contribution:** 2
**Rating:** 2
**Confidence:** 3

**Summary:**

This paper introduces a unifying framework for gradient aggregation in multi-objective optimization, establishing convergence rates to Pareto stationarity by defining a sufficient alignment condition and providing a primal optimization perspective that unifies and clarifies the theoretical relationships between existing algorithms.

**Strengths:**

1. This paper attempts to propose a unified framework.
2. Supported by extensive proof combining theory and experiments.

**Weaknesses:**

1. CAPPED MGDA not compared in Table 2?
2. Figure 4 does't include comparisons with other methods?
3. This framework may only be applicable to all current methods.

**Questions:**

N/A

---

> ### Author Response · Authors · 2025-11-21
> **Thank you for your feedback. We address your questions below and clarify several points regarding our results and the scope of the framework.**
>
> - **Capped MGDA not compared in Table 2?**
>
>   Our experiments are structured to illustrate and validate the theoretical results developed in the paper. In particular, Table 2 focuses on __non-conflicting__ aggregation methods—those that fall within the scope of Theorem 2. Capped-MGDA does not satisfy the non-conflicting condition in general and is therefore excluded from this comparison by design.
>
>   Instead, Capped-MGDA is evaluated separately in Section 5.2, where we examine its intended use case (adversarial FL) and its interpolation behavior between MGDA and linear scalarization. In the revision, we further strengthen this section by (1) adding results for a wider range of capping values C, and (2) including a plot of the non-conflictingness measure across iterations, which explicitly shows that Capped-MGDA can violate non-conflictingness and further justifies why it does not appear in Table 2.
>
> - **Figure 4 doesn’t include comparisons with other methods?**
>
>   Figure 4 focuses on the adversarial FL setting and compares MGDA with Capped-MGDA, as these two methods are closely related, and this adversarial setting is a primary motivation for introducing Capped-MGDA. The purpose of this figure is to illustrate the improved robustness of Capped-MGDA under adversarial gradients and to show the effect of the capping parameter C, rather than to repeat the broader comparisons among baselines already presented in Section 5.1.
>
>   Thanks for your suggestion, for completeness, we have added final average accuracy comparisons against other relevant methods in Table 4, Appendix C. These results complement Figure 4 by showing that Capped-MGDA performs competitively relative to a wider set of baselines in the same adversarial FL setup.
>
> - **Framework may only be applicable to all current methods?**
>
>   This comment seems to be a slight misunderstanding of our work. Our aim is indeed to unify existing gradient aggregation methods under shared theoretical principles (e.g., MGDA, Nash-MTL, UPGrad, CAGrad), but the framework is __not limited to__ these methods.
>
>   For example, Theorem 2 applies to all methods whose convex aggregated direction is non-conflicting. Theorem 4 applies to all methods whose subproblem construction builds on a decreasing and convex $s$ and $r \geq \frac{1}{2}\|\cdot\|^2$, which includes power-mean-based methods (e.g. Nash-MTL, MGDA) and convex-risk-measure-based ones (e.g., capped MGDA). In particular, both Capped-MGDA and GreedyGCP presented in the paper (see Appendix A.2) are __new__ methods developed directly from our framework. This demonstrates that the theoretical conditions in Theorem 2 and Theorem 4 are sufficiently broad to inspire and support the design of new approaches, in addition to unifying existing ones.

---

> ### Author Response · Authors · 2025-11-27
> **Follow-Up on Rebuttal Discussion**
>
> Dear Reviewer agTX,
>
> As the discussion period is nearing its end, we would like to kindly follow up to see if you have any further feedback, especially regarding our clarifications on Table 2 and Figure 4, as well as the scope of our proposed framework. We are happy to elaborate further if any part of our response remains unclear.
>
> Thank you again for your time and for reviewing our paper.
>
> Best regards,
> The Authors

---

### Official Review · Reviewer_HyR5 · 2025-11-03

**Soundness:** 2
**Presentation:** 2
**Contribution:** 2
**Rating:** 4
**Confidence:** 2

**Summary:**

This paper introduces a unifying convergence analysis for multi-objective optimization (MOO) algorithms. Specifically, it introduces a set of general conditions and results based on the well-known feasible direction lemmas. Then it shows how non-conflicting gradient condition satisfies the general result. The unifying framework is able to provide convenient analysis for a large family of MOO methods, including new ones proposed by this paper. Experiments on a synthetic task and federated image classification are provided to support the result.

**Strengths:**

1. This paper is well-written and relatively easy to follow.
2. The unifying analysis provides a convenient check for new MOO updates.

**Weaknesses:**

My main concern on this paper is on its theoretical originality/novelty and its experimental soundness. The main theoretical results are not new, and the newly proposed algorithms are not sufficiently tested with experiments. As an example, the convergence analysis of Theorem 1 (which is the key result for the theories in this paper) is very similar to that of the single-objective case. The new algorithm Capped-MGDA is tested in a federated image classification task under adversarial attacks, which might be too toy to verify its broad effectiveness., e.g., in normal settings with no attacks or federated aggregations.

**Questions:**

In line 120--121, the author argued that non-conflicting condition is not merely a preference, but a fundamental condition for convergence to a Pareto stationary point. This is later supported by Theorem 2, which shows that non-conflicting condition leads to the convergence result to Pareto stationary points (Corollary 1). However, even optimizing an arbitrary convex combination of the smooth objectives lead to convergence to Pareto stationary points. In this sense, why is non-conflicting aggregation crucial and how does Theorem 2 support it?

**Details Of Ethics Concerns:**

No ethics concerns.

---

> ### Author Response · Authors · 2025-11-21
> **Thank you for the critical feedback. We address your concerns below, with particular attention to questions regarding the novelty of our contributions.**
>
> - **Main Concern: Theoretical originality/novelty. For example, the convergence analysis of Theorem 1 is similar to that of the single-objective case.**
>
>   Thank you for raising this concern. We appreciate this chance to further clarify the role of Theorem 1 and the overall contribution of our theoretical framework:
>
>   __1. The big picture of our main theoretical contributions.__
>   Theorem 1 is intentionally presented in a form resembling the classical single-objective descent condition. This is by design, because one of our goals is to build a new _bridge_ between single-objective optimization and multi-objective optimization (MOO). The novelty and contributions of the paper do not lie in Theorem 1 alone, but also in how it __enables__ the later results. The new theoretical contributions emerge in:
>   - Theorem 1 (new): proposes a general _sufficient alignment condition_ that is especially useful for the analysis of gradient aggregation methods in the __multi-objective setting__.
>
>   - Corollary 1 (new): connects convergence in $\|\mathbf{d}_t\|$ to convergence in Pareto-stationarity ($\gamma(\mathbf{w}_t)$). Based on Theorem 1.
>
>   - Theorem 2 (new): establishes a unified convergence result for __all non-conflicting aggregation methods__, including MGDA, Nash-MTL, UPGrad, and others. Based on Theorem 1.
>
>   - Theorem 3 (new): stronger guarantee for non-conflicting MOO methods in the convex case.
>
>   - Theorem 4 (new): provides convergence guarantees for a broad class of __subproblem-based aggregation methods__, including capped MGDA and many others. Based on Theorem 1.
>
>   These results have not appeared in prior literature, especially not in a unified form applicable across modern multi-objective gradient-based methods.
>
>   __2. It is actually _good_ that Theorem 1 resembles the single-objective case.__
>
>   The resemblance is elegant and conceptually important:
>   - It shows that __modern multi-objective gradient aggregation algorithms can be analyzed within a simple, general framework__, much like single-objective methods.
>   - It collapses many previously complicated, method-specific arguments into a __single clean condition__ (Condition (A)).
>   - It reveals that the core requirement is a __generalized__ alignment condition rather than the diverse assumptions used in existing work.
>
>   Thus, the similarity to single-objective case is not a lack of novelty; it is precisely what makes the framework powerful and broadly applicable in the multi-objective setting.
>
>   __3. The unification is new: many existing methods are incorporated for the first time.__
>
>   To our knowledge, this paper provides the first __unified__ convergence analysis covering (not limited to): Linear Scalarization, MGDA, Nash-MTL, UPGrad, PCGrad, CAGrad, DualProj, FairGrad, (our new) Capped-MGDA, and even Mixed-aggregator scheduling.
>   Previously, each method had its own __specialized__ convergence proof. Our framework covers all of these under the same theoretical lens.
>
>   __4.The novelty is in the _framework_, not in the proof of Theorem 1 itself.__
>
>   Theorem 1 is the _starting point_, not the endpoint. Its purpose is to provide a simple descent inequality using a surrogate function $F$. We also point out that $F$ is introduced solely to facilitate the analysis, it does not imply the algorithm should explicitly optimize it (as in the case of fixed linear scalarization). This decoupling between the __actual algorithm__ and the __surrogate function__ is one of the conceptual novelties of our work, and it is what makes the unification possible.

---

> ### Author Response · Authors · 2025-11-21
>
> - **Why Capped-MGDA is tested in the adversarial FL setting.**
>
>   Capped-MGDA is derived from a primal CVaR-style risk formulation and leads to an MGDA-like dual QP problem, with the key difference that the coefficients $\lambda$ are constrained by an upper cap. As reviewer gtaF correctly noted, the CVaR formulation is expected to yield greater robustness against adversarial gradients compared to standard MGDA. For this reason, the adversarial FL setting—where CIFAR-10 is a widely used benchmark—provides a more representative and meaningful environment to evaluate and validate the intended theoretical advantages of Capped-MGDA. Conducting the experiments in this setting therefore __offers a clearer demonstration of the motivation__ behind the method.
>
>   And thanks to your suggestion, we have __added additional results__ in the standard (non-adversarial) setting (see Figure 4, top right) for a more comprehensive comparison. We observe that MGDA and Capped-MGDA perform similarly when no attacks are present, whereas MGDA becomes significantly more vulnerable under adversarial gradients. Taken together with the results in Table 4 and Figure 8, these observations suggest that Capped-MGDA is an effective and more robust alternative to MGDA.
>
>    Also, to strengthen our empirical evaluation, we have added comparisons across a range of capping values C, as well as a figure visualizing the "non-conflictingness" of $\mathbf{d}_t$ throughout optimization [see Figure 4 (Bottom Right)]. The results show a clear trade-off between non-conflictingness and update norm: bigger values of $C$ make the method behave more like MGDA—yielding stronger non-conflictingness but smaller update norms, which lead to weaker final performance. In contrast, smaller values of $C$ behave more like linear scalarization, producing larger update norms but exhibiting reduced non-conflictingness.

---

> ### Author Response · Authors · 2025-11-21
>
> - **Why is non-conflicting aggregation crucial and how does Theorem 2 support it? Why not just optimize an arbitrary convex combination?**
>
>   Thank you for this question. Pareto stationarity (PS) and non-conflicting aggregation play different roles: PS is a property of a solution $\mathbf{w}_*$, whereas non-conflictingness is a per-iteration property of the update direction $\mathbf{d}_t$:
>   $$\left\langle\mathbf{d}_t, \nabla f_k\left(\mathbf{w}_t\right)\right\rangle \geq 0, \quad \forall k $$
>   A method that converges to a PS point does __not__, in general, have to be non-conflicting at every step. The contribution of our work is to show that, __in the other direction__, non-conflictingness is in fact a very powerful and unifying condition that implies convergence to PS. Concretely, Theorem 2 proves that any method whose convex aggregated direction is non-conflicting must satisfy condition (A), which in turn implies convergence toward Pareto stationarity (by Corollary 1). To our knowledge, such a general 'non-conflicting => PS convergence' result did not exist before.
>
>   Meanwhile, non-conflicting aggregation is of __independent interest__ in the related literature and is often regarded as a desirable property in its own right, motivating the development of several new methods designed specifically to satisfy it (e.g., [1,2,3]). Intuitively, non-conflictingness ensures that each update is locally improving for _all_ objectives, which is particularly important in applications such as fairness [4], where degrading any objective can be unacceptable. It also allows one to __refine__ a given candidate $\mathbf{w}$ using a non-conflicting direction without harming any objective, something that cannot be guranteed when optimizing a fixed convex combination of the objectives; we refer to the reference [5] for more discussions on the limitations of optimizing a fixed convex combination of objectives.
>
> [1] Hwang, Y., & Lim, D. (2024). Dual cone gradient descent for training physics-informed neural networks. _Advances in Neural Information Processing Systems_, 37, 98563-98595.
>
> [2] Quinton, P., & Rey, V. (2024). Jacobian descent for multi-objective optimization. _arXiv preprint_ arXiv:2406.16232.
>
> [3] Liu, Q., Chu, M., & Thuerey, N. (2024). Config: Towards conflict-free training of physics informed neural networks. _ICLR 2025._
>
> [4] Hu, Z., Shaloudegi, K., Zhang, G., & Yu, Y. (2022). Federated learning meets multi-objective optimization. _IEEE Transactions on Network Science and Engineering_, 9(4), 2039-2051.
>
> [5] Hu, Y., Xian, R., Wu, Q., Fan, Q., Yin, L., & Zhao, H. (2023). Revisiting scalarization in multi-task learning: A theoretical perspective. _Advances in Neural Information Processing Systems_, 36, 48510-48533.

---

> ### Author Response · Authors · 2025-11-27
> **Follow-Up on Rebuttal Discussion**
>
> Dear Reviewer HyR5,
>
> As the discussion period is nearing its end, we would like to kindly follow up to see if you have any additional feedback, particularly regarding our clarifications on the theoretical originality of our results and the role of non-conflicting aggregation. We are happy to elaborate further if anything remains unclear.
>
> Thank you again for your time and thoughtful review.
>
> Best regards,
> The Authors

---

### Author Response · Authors · 2025-11-25

We thank all reviewers for their thoughtful and constructive feedback. In the individual responses, we have addressed each question and concern in detail. Based on the reviewers’ comments, we realized that certain aspects of the theoretical framework could benefit from clearer exposition. We have therefore refined the presentation in the main paper to better highlight how the key ideas fit together and to improve the readability of the main results. We also polished *Figure 1* to provide a clearer visual roadmap of the core concepts and their relationships, which we hope will help readers quickly grasp the structure of our framework upon a brief reread.



Below we summarize the main improvements and clarifications made in the revised manuscript (highlighted in blue):

  - 1. Expanded discussion in Section 5.1 to clarify the convergence behaviors in Figures 2 and 3, explaining how the empirical trends support Theorem 2, addressing questions from Reviewers MKzs and gatF.

  - 2. Expanded Section 5.2 with a more comprehensive study of capped MGDA in the (adversarial) FL setting:
    - added experiments with multiple values of the capping coefficient $C$ to illustrate its effect (Reviewer gatF);
    - added the standard (no-adversary) setting as a baseline for comparison (Reviewer HyR5);
    - included a plot measuring the degree of non-conflictingness;
    - added comparisons with additional baseline methods (Reviewer gatF and agTX).

  - 3. Improved the readability of the theoretical results by polishing the theorem statements (e.g., adding theorem names and clarifying their presentation), making them easier to follow in the revised version.

  - 4. Added detailed discussions in Appendix A.1 on the convergence guarantees of many existing methods, including FairGrad and PIVRG, and clarified how our framework applies to, and in some cases strengthens, these guarantees.

  - 5. Improved notation consistency and added further justification in the proofs to address Reviewer MKzs’s concerns.

We believe the revisions greatly improve the clarity and accessibility of the paper, particularly in understanding the structure and implications of our theoretical framework. The additional experiments and expanded discussions directly address the reviewers’ concerns and further validate the effectiveness and generality of our approach. We hope that these improvements will help readers more clearly see the contributions and overall value of our work in the revised version.

---

### Meta-Review · Area_Chair_YXYM · 2025-12-15

**Summary:**

This paper presents a unified convergence analysis for a class of non-conflicting multi-objective optimization (MOO) algorithms, including MGDA, Nash-MTL, and CAGrad. The analysis builds on an existing theorem characterizing feasible descent directions under a general non-conflicting condition at each iteration (Eq. (9)). Based on this result, the authors derive several corollaries and propositions establishing convergence guarantees in both nonconvex and convex settings. They further demonstrate that existing non-conflicting conditions used in prior MOO algorithms satisfy a key requirement (Condition (A)), thereby ensuring convergence to Pareto-stationary solutions. Finally, the paper extends the framework to propose two new algorithms, Capped-MGDA (under a risk measure) and Greedy-DCP.

**Reviewer Concerns:**

All reviewers agree that the paper provides a general analytical framework capable of establishing convergence for a broad class of MOO algorithms. However, several concerns were raised. Reviewer HyR5 questioned the technical novelty and experimental soundness, noting that the main convergence result (Theorem 1) closely resembles its single-objective counterpart and that experiments for the new Capped-MGDA algorithm are limited to a specific adversarial federated learning setting. Reviewer MKzs raised technical questions regarding the derivations and asked whether the theoretical analysis extends to the stochastic case. Reviewer gatF gave a positive evaluation overall but requested clarifications and offered suggestions for improving the experimental section. The authors provide a thorough response to these comments, including additional experiments (e.g., more baselines in federated learning setups and results in non-adversarial settings) as well as clarifications and expanded discussions of the convergence results and notation. However, several concerns regarding technical novelty and the significance of the results are still outstanding.

After reading the paper by myself, going through the reviews, and the authors’ responses, I believe that providing a unified convergence analysis for non-conflicting MOO algorithms is a meaningful contribution. The framework offers a useful recipe for designing new provably convergent MOO algorithms, as illustrated by Capped-MGDA. That said, some concerns remain. First, the technical novelty may be limited, as noted by Reviewer HyR5, since the main results heavily rely on the established Theorem 1. Under the non-conflicting condition, which ensures descent-like behavior across multiple objectives, the analysis itself appears relatively straightforward. In my view, a key contribution lies in demonstrating that existing algorithms, particularly Nash-MTL-type methods, satisfy the required conditions. Notably, while the original Nash-MTL paper assumes that the final solution lies in the convex hull of gradients, this paper establishes the result via dual derivations. However, even in this case, some technical questions arise. For Nash-MTL-type algorithms, the input x of function s(x) cannot be zero, whereas the analysis in Section 4.2 assumes s(0)=r(0)=0, which may not directly apply. This discrepancy should be carefully revisited. Moreover, in practice, the subproblems (e.g., Eq. (12)) cannot be solved exactly, leading to optimization errors. It remains unclear how such errors affect the convergence guarantees and overall complexity, particularly under different algorithmic structures (single-loop versus double-loop). Recent works, such as [1], have devoted significant effort to addressing these issues in single-loop settings. Finally, extending the analysis or discussion to stochastic settings would further strengthen the paper.

[1] Fernando, H. et al. (2023). "Mitigating Gradient Bias in Multi-objective Learning: A Provably Convergent Approach"

**Reviewer Scores:**

None of the reviewers interacted with the authors during the discussion period. Reviewer HyR5’s main concern centers on the technical novelty of the paper, which I find reasonable after carefully reading the manuscript. As a result, it is unlikely that this reviewer’s score would increase. Reviewer agTX provided the shortest review and offered limited substantive feedback; consequently, I did not place significant weight on these comments. Reviewer MKzs raised several reasonable questions. In particular, requests for clarification regarding the derivations were, in my opinion, addressed satisfactorily in the authors’ response. However, the concern regarding the stochastic setting remains valid and was not explicitly resolved with concrete solutions. Therefore, this reviewer’s score is also unlikely to increase.

Overall, I appreciate the unified perspective provided in this paper; however, it does not appear sufficient for acceptance at this time. With strengthened theoretical analysis and a more comprehensive experimental evaluation, the paper has the potential to become a strong submission to a future top machine learning conference.

---

### Decision · Program_Chairs · 2026-01-26

Reject